# TEST-TIME ADAPTATION VIA SELF-TRAINING WITH NEAREST NEIGHBOR INFORMATION

**Minguk Jang,   Sae-Young Chung,   Hye Won Chung**
School of Electrical Engineering
Korea Advanced Institute of Science and Technology (KAIST)
Daejeon, Republic of Korea
{mgjang, schung, hwchung}@kaist.ac.kr

## ABSTRACT

Test-time adaptation (TTA) aims to adapt a trained classifier using online unlabeled test data only, without any information related to the training procedure. Most existing TTA methods adapt the trained classifier using the classifier's prediction on the test data as pseudo-label. However, under test-time domain shift, accuracy of the pseudo labels cannot be guaranteed, and thus the TTA methods often encounter performance degradation at the adapted classifier. To overcome this limitation, we propose a novel test-time adaptation method, called *Test-time Adaptation via Self-Training with nearest neighbor information (TAST)*, which is composed of the following procedures: (1) adds trainable adaptation modules on top of the trained feature extractor; (2) newly defines a pseudo-label distribution for the test data by using the nearest neighbor information; (3) trains these modules only a few times during test time to match the nearest neighbor-based pseudo label distribution and a prototype-based class distribution for the test data; and (4) predicts the label of test data using the average predicted class distribution from these modules. The pseudo-label generation is based on the basic intuition that a test data and its nearest neighbor in the embedding space are likely to share the same label under the domain shift. By utilizing multiple randomly initialized adaptation modules, TAST extracts useful information for the classification of the test data under the domain shift, using the nearest neighbor information. TAST showed better performance than the state-of-the-art TTA methods on two standard benchmark tasks, domain generalization, namely VLCS, PACS, OfficeHome, and TerraIncognita, and image corruption, particularly CIFAR-10/100C. Our code is available at https://github.com/mingukjang/TAST.

## 1 INTRODUCTION

Deep neural networks often encounter significant performance degradations under domain shift (i.e., distribution shift). This phenomenon has been observed in various tasks including classification (Taori et al., 2020; Wang et al., 2021b), visual recognition (Saenko et al., 2010; Csurka, 2017), and reinforcement learning (Cobbe et al., 2019; Mendonca et al., 2020; Lee and Chung, 2021b). There are two broad classes of domain adaptation methods that attempt to solve this problem: supervised domain adaptation (SDA) (Tzeng et al., 2015; Motiian et al., 2017) and unsupervised domain adaptation (UDA) (Ganin and Lempitsky, 2015; Long et al., 2016; Sener et al., 2016). Both SDA and UDA methods aim to obtain domain-invariant representations by aligning the representations of training and test data closely in the embedding space. While testing, UDA methods require the training dataset and SDA methods additionally require labeled data of the test domain. However, in practice, it is often difficult to access training datasets or labeled data in the test domain during test time, due to data security or labeling cost.

Test-time adaptation (TTA) (Iwasawa and Matsuo, 2021; Wang et al., 2021a) is a prominent approach to alleviate the problems caused by the domain shift. TTA methods aim to adapt the trained model to the test domain without a labeled dataset in the test domain and any information related to the training procedure (e.g., training dataset, feature statistics of training domain (Sun et al., 2020;

Liu et al., 2021; Eastwood et al., 2022)). TTA methods have access to the online unlabeled test data only, whereas domain adaptation methods assume access to the whole (i.e., offline) test data.

There are three popular categories for TTA: normalization-based method (Schneider et al., 2020), entropy minimization (Liang et al., 2020; Wang et al., 2021a) and prototype-based methods (Iwasawa and Matsuo, 2021). Normalization method replaces the batch normalization (BN) statistics of the trained model with the BN statistics estimated on test data, and does not update model parameters except for the BN layers. Entropy minimization methods fine-tune the trained feature extractor, which is the trained classifier except the last linear layer, by minimizing the prediction entropy of test data. These methods force the classifier to have over-confident predictions for the test data, and thus have a risk of degrading model calibration (Guo et al., 2017; Mukhoti et al., 2020), a measure of model interpretability and reliability. One form of entropy minimization is self-training (Rosenberg et al., 2005; Lee, 2013; Xie et al., 2020). Self-training methods use predictions from the classifier as pseudo labels for the test data and fine-tune the classifier to make it fit to the pseudo labels. These methods have a limitation that the fine-tuned classifier can overfit to the inaccurate pseudo labels, resulting in confirmation bias (Arazo et al., 2020). This limitation can be harmful when the performance of the trained classifier is significantly degraded due to the domain shift. On the other hand, Iwasawa and Matsuo (2021) proposed a prototype-based TTA method, named T3A, that simply modifies a trained linear classifier (the last layer) by using the pseudo-prototype representations of each class and the prototype-based classification for test data, where the prototypes are constructed by previous test data and the prediction for the data from trained classifier. T3A does not update the trained feature extractor at test time. T3A is simple but it brings a marginal performance gain (Table 1 and 3).

In this work, we propose a new test-time adaptation method, which is simple yet effective in mitigating the confirmation bias problem of self-training, by adding adaptation modules on top of the feature extractor, which are simply trainable during test time. We use the prototype-based classifier as in T3A, but not in the embedding space of the original feature extractor but in the embedding space of the adaptation modules, trained with nearest neighbor information, to achieve higher performance gains than the original simple prototype-based classifier method. Our method, named *Test-time Adaptation via Self-Training with nearest neighbor information (TAST)*, is composed of the following procedures: (1) adds randomly initialized adaptation modules on top of the feature extractor at the beginning of test time (Figure 1); (2) generates pseudo label distribution for a test data considering the nearest neighbor information; (3) trains the adaptation modules only a few times during test time to match the nearest neighbor-based pseudo label distribution and a prototype-based class distribution for the test data; and (4) predicts the label of test data using the average predicted class distribution from the adaptation modules. Specifically, in (1), we add the trainable adaptation modules to obtain new feature embeddings that are useful for classification in the test domain. In (2), TAST assigns the mean of the labels of the nearby examples in the embedding space as the pseudo label distribution for the test data based on the idea that a test data and its nearest neighbors are more likely to have the same label. In (3), TAST trains the adaptation modules to output the pseudo label distribution when the test data is fed into (Figure 1 *Right*). And in (4), we average the predicted class distributions from adaptation modules for the prediction of test data (Figure 1 *Left*).

We investigate the effectiveness of TAST on two standard benchmarks, domain generalization and image corruption. We demonstrate that TAST outperforms the current state-of-the-art test-time adaptation methods such as Tent (Wang et al., 2021a), T3A, and TTT++ (Liu et al., 2021) on the two benchmarks. For example, TAST surpasses the current state-of-the-art algorithm by $1.01\%$ on average with ResNet-18 learned by Empirical Risk Minimization (ERM) on the domain generalization benchmarks. Extensive ablation studies show that both the nearest neighbor information and the adaptation module utilization contribute to the performance increase. Moreover, we experimentally found that the adaptation modules adapt feature extractor outputs effectively although the adaptation modules are randomly initialized at the beginning of test time and trained with a few gradient steps per test batch during test time.

## 2    PRELIMINARIES

**Test-time domain shift**    Consider a labeled dataset $D^{\text{train}} = \{(x_i, y_i)\}_{i=1}^{n_{\text{train}}}$ drawn from a distribution $P^{\text{train}}$, where $x \in \mathbb{R}^d$ and $y \in \mathcal{Y} := \{1, 2, \cdots, K\}$ for a $K$-class classification problem.

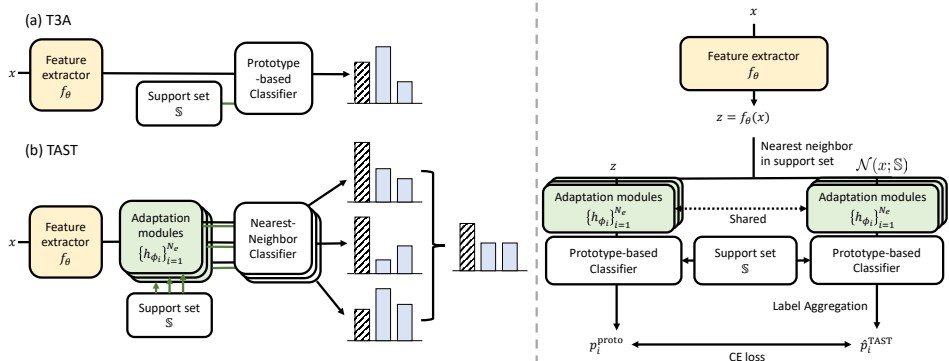

Figure 1: Overview of TAST. *Left:* A schematic of TAST compared to T3A. The dashed class indicates the ground-truth class. (a) T3A constructs prototypes that represent classes in the embedding space of feature extractor $f_\theta$ using a support set $\mathbb{S}$. Then T3A predicts the label of the test data $x$ as the class of the nearest prototype. (b) TAST adds trainable adaptation modules $\{h_{\phi_i}\}$ on top of $f_\theta$ and computes the estimated class distributions of $x$ by aggregating the pseudo labels of the nearest support examples of $x$ in the embedding space of adaptation modules. *Right:* Overview of TAST training. Based on the intuition that a test data $x$ and its nearest neighbors $\mathcal{N}(x; \mathbb{S})$ are likely to share the same label, we use the mean of prototype-based predictions of the support examples in $\mathcal{N}(x; \mathbb{S})$ as the pseudo label of $x$. We train the adaptation modules to predict the pseudo labels when the test data is fed into. Notice that the feature extractor $f_\theta$ is frozen during test time.

A number of classifiers have been proposed that easily classify unseen test data under the i.i.d. assumption that unseen test data $D^{\text{test}}$ is drawn from the same distribution as training data, i.e., $P^{\text{train}} = P^{\text{test}}$. We assume the classifier is a deep neural network composed of two parts: a feature extractor $f_\theta : \mathbb{R}^d \rightarrow \mathbb{R}^{d_z}$ and a linear classifier $g_w : \mathbb{R}^{d_z} \rightarrow \mathcal{Y}$, where $\theta$ and $w$ are the neural network parameters. ERM optimizes $\theta$ and $w$ to obtain a good classifier for future samples in $D^{\text{test}}$ by minimizing the objective function $\mathcal{L}(\theta, w) = \mathbb{E}_{(x,y) \in D^{\text{train}}} [l(g_w(f_\theta(x)), y)]$, where $l$ is a loss function such as cross-entropy loss. However, under the test-time domain shift (i.e., distribution shift), the i.i.d. assumption between the training and test distributions does not hold, i.e., $P^{\text{train}} \neq P^{\text{test}}$, and the trained classifiers often show poor classification performance for the test data.

**Prototype-based classification in test-time adaptation**   Prototype-based classification refers to a method that obtains prototype representations, which represent each class in the embedding space, and then predicts the label of an input as the class of the nearest prototype. Since labeled data is not available in the TTA setting, T3A (Iwasawa and Matsuo, 2021) utilizes a support set that is composed of previous test data and their predictions for the test data by the trained classifier. T3A does not modify parameters of the classifier. Since the embedding space of the feature extractor is unchanged during the test time, T3A constructs the support set using the feature representations for test data instead of the data itself. Specifically, a support set $\mathbb{S}_t = \{\mathbb{S}_t^1, \mathbb{S}_t^2, \cdots, \mathbb{S}_t^K\}$ is a set of test samples until time $t$. The support set is initialized with the weight of the last linear classifier, i.e., $\mathbb{S}_0^k = \left\{ \frac{w^k}{\|w^k\|} \right\}$, where $w^k$ is the parts of $w$ related to $k$-th class for $k = 1, 2, \ldots, K$. At time $t$, the support set is updated as

$$\mathbb{S}_t^k = \begin{cases} \mathbb{S}_{t-1}^k \cup \left\{ \frac{f_\theta(x_t)}{\|f_\theta(x_t)\|} \right\} & \text{if } \arg\max_c p_c = k \\ \mathbb{S}_{t-1}^k & \text{otherwise,} \end{cases} \tag{1}$$

where $p_k$ represents the likelihood that the classifier assigns $x_t$ to the $k$-th class. Using the support set $\mathbb{S}_t^k$, one can obtain the class prototype for class $k$ by taking the centroid of the representations in the support set. Formally, the prototype $\mu_k$ for class $k$ is computed as $\mu_k = \frac{1}{|\mathbb{S}_t^k|} \sum_{z \in \mathbb{S}_t^k} z$ for $k = 1, 2, \cdots, K$. Then, the prediction for an input $x_t$ is made by comparing the distances between the embedding of $x_t$ and the prototypes, i.e., $\hat{y} = \arg\min_c d(f_\theta(x_t), \mu_c)$ with a pre-defined metric $d$ such as Euclidean distance or cosine similarity.[1] Since the wrongly pseudo labeled examples can degrade the classification performance, the support examples with unconfident pseudo labels are regarded as unreliable examples and filtered out, i.e., at time stamp $t$,

---

[1] We use the cosine similarity as a distance metric $d$ for experiments throughout this paper.

---

**Algorithm 1** Test-time Adaptation via Self-Training with nearest neighbor information (TAST)

---

**Require:** Feature extractor $f_\theta$, number of adaptation modules $N_e$, adaptation modules $\{h_{\phi_i}\}_{i=1}^{N_e}$, test batch $\mathbb{B}$, support set $\mathbb{S}$, number of gradient steps per adaptation $T$, number of support examples per each class $M$, number of nearby support examples $N_s$, learning rate $\alpha$
**Ensure:** Predictions for all $x \in \mathbb{B}$
   Update the support set $\mathbb{S}$ with eq. (1) and the entropy-based filtering
   Retrieve the nearest neighbors $\mathcal{N}(x; \mathbb{S})$ for all $x \in \mathbb{B}$ with eq. (2)
   **for** $t = 1 : T$ **do**
      **for** $i = 1 : N_e$ **do**
         **for** $x \in \mathbb{B}$ **do**
            Obtain the nearest neighbor-based pseudo label $\hat{p}_i^{\text{TAST}}(\cdot|x)$ of $x$ with eq. (4)
            Compute the prototype-based class distribution $p_i^{\text{proto}}(\cdot|x)$ of $x$ with eq. (6)
         **end for**
         $\phi_i \leftarrow \phi_i - \alpha \nabla_{\phi_i} \frac{1}{|\mathbb{B}|} \sum_{x \in \mathbb{B}} \text{CE}(\hat{p}_i^{\text{TAST}}(\cdot|x), p_i^{\text{proto}}(\cdot|x))$
      **end for**
   **end for**
   Compute the predictions for all $x \in \mathbb{B}$ with eq. (8)

---

$\mathbb{S}_t^k \leftarrow \{z | z \in \mathbb{S}_t^k, H(\sigma(g_w(z))) \le \alpha_k\}$, where $\alpha_k$ is the $M$-th largest prediction entropy of the samples from $\mathbb{S}_t^k$, $H$ is Shannon entropy (Lin, 1991), and $\sigma$ is the softmax function. T3A modifies only the support set configuration and does not update the trained model parameters at test time. Thus, T3A cannot effectively mitigate the classification performance degradation caused by test-time domain shift. To address this issue, we extract useful information for classification of the test data by utilizing multiple randomly initialized adaptation modules that are trained using nearest neighbor-based pseudo labels.

## 3 METHODOLOGY

In this section, we describe two main components of our method TAST: adaptation module utilization (Section 3.1) and pseudo-label generation considering nearest neighbor information (Section 3.2).

### 3.1 ADAPTATION MODULE

We first discuss the parts to be fine-tuned in the trained classifier before explaining our test-time adaptation method. One possible choice is to fine-tune the whole network parameters in the classifier during test time, but this approach can be unstable and inefficient (Wang et al., 2021a; Kumar et al., 2022). Another choice is to fine-tune only the parameters of batch normalization (BN) layers in the classifier as in Wang et al. (2021a). Although it achieves effective test-time adaptation, it has a limitation that it can be utilized only if there are BN layers in the trained classifier. The other choice is to train a new classifier added on top of the frozen feature extractor during test time as in Lee and Chung (2021a). We construct the new classifier by adding a randomly initialized adaptation module as illustrated in Figure 1. During the test time, we train the adaptation module and predict the label of the test data using prototype-based class distributions from the adaptation module. The random initialization of the adaptation module may cause performance degradation of trained classifier. Thus, we consider an ensemble scheme (Wen et al., 2020; YM. et al., 2020; Mesbah et al., 2021) to alleviate the issues caused by the random initialization of the adaptation modules to obtain more robust and accurate predictions. We train the adaptation modules independently and predict the label of the test data using the average predicted class distribution from the adaptation modules.

### 3.2 SELF-TRAINING WITH NEAREST NEIGHBOR INFORMATION

TAST generates pseudo label distributions for unlabeled test data with the nearest neighbor information and fine-tunes the adaptation modules with the pseudo label distributions. The whole adaptation procedure of TAST is described in Algorithm 1. We first update the support set $\mathbb{S}$ and filter out the unconfident examples from the support set as in Iwasawa and Matsuo (2021). Then, we find $N_s$ nearby support examples of test data $x$ in the embedding space of $f_\theta$. We denote $\mathcal{N}(x; \mathbb{S})$ as the set

of nearby support examples of $x$,

$$\mathcal{N}(x;\mathbb{S}) := \{z \in \mathbb{S} | d(f_\theta(x), z) \leq \beta_x\}, \tag{2}$$

where $\beta_x$ is the distance between $x$ and the $N_s$-th nearest neighbor of $x$ from $\mathbb{S}$ in the embedding space of $f_\theta$. Each adaptation module is trained individually during test time. For the $i$-th adaptation module $h_{\phi_i}$[2], we compute the prototype representations $\mu_{i,1}, \mu_{i,2}, \ldots, \mu_{i,K}$ in the embedding space of $h_{\phi_i} \circ f_\theta$ with a support set $\mathbb{S} = \{\mathbb{S}^1, \mathbb{S}^2, \cdots, \mathbb{S}^K\}$, i.e., $\mu_{i,k} = \frac{1}{|\mathbb{S}^k|} \sum_{z \in \mathbb{S}^k} h_{\phi_i}(z)$, for $k = 1, 2, \ldots, K$. With the prototypes, we compute the prototype-based predicted class distribution of the nearby support examples in the embedding space of $h_{\phi_i} \circ f_\theta$, i.e., for $z \in \mathcal{N}(x;\mathbb{S})$, the likelihood that the prototype-based classifier assigns $z$ to the $k$-th class is computed as

$$p_i^{\text{proto}}(k|z) := \frac{\exp(-d(h_{\phi_i}(z), \mu_{i,k})/\tau)}{\sum_c \exp(-d(h_{\phi_i}(z), \mu_{i,c})/\tau)}, \tag{3}$$

where $\tau$ is the softmax temperature[3]. With the nearest neighbor information, TAST generates a pseudo label distribution $\hat{p}_i^{\text{TAST}}$ of $x$ by aggregating prototype-based predicted class distribution of the nearby support examples in $\mathcal{N}(x;\mathbb{S})$ as

$$\hat{p}_i^{\text{TAST}}(k|x) := \frac{1}{N_s} \sum_{z \in \mathcal{N}(x;\mathbb{S})} \mathbb{1}[\arg\max_c p_i^{\text{proto}}(c|z) = k], \tag{4}$$

for $k = 1, 2, \ldots, K$. Specifically, we use the one-hot class distributions for pseudo label generation as in Lee (2013); Sohn et al. (2020). Then, we fine-tune the adaptation modules by minimizing the cross-entropy loss between the predicted class distribution of the test example and the nearest neighbor-based pseudo label distribution:

$$\mathcal{L}^{\text{TAST}}(\phi_i) = \frac{1}{|D^{\text{test}}|} \sum_{x \in D^{\text{test}}} \text{CE}(\hat{p}_i^{\text{TAST}}(\cdot|x), p_i^{\text{proto}}(\cdot|x)), \tag{5}$$

$$p_i^{\text{proto}}(k|x) := \frac{\exp(-d(h_{\phi_i}(f_\theta(x)), \mu_{i,k})/\tau)}{\sum_c \exp(-d(h_{\phi_i}(f_\theta(x)), \mu_{i,c})/\tau)}, \ k = 1, 2, \ldots, K, \tag{6}$$

where CE denotes the standard cross-entropy loss. We iterate the pseudo labeling and fine-tuning processes for $T$ steps per batch. We note that our method does not propagate gradients into the pseudo labels as in Laine and Aila (2017); Berthelot et al. (2019). Finally, we predict the label of $x$ using the average predicted class distribution $p_i^{\text{TAST}}$ from the adaptation modules, i.e.,

$$p_i^{\text{TAST}}(k|x) := \frac{1}{N_s} \sum_{z \in \mathcal{N}(x;\mathbb{S})} p_i^{\text{proto}}(k|z) \tag{7}$$

$$\hat{y}^{\text{TAST}} = \arg\max_c p^{\text{TAST}}(c|x) = \arg\max_c \frac{1}{N_e} \sum_{i=1}^{N_e} p_i^{\text{TAST}}(c|x) \tag{8}$$

Additionally, we consider a variant of TAST, named TAST-BN, that fine-tunes the BN layers instead of adaptation modules. The support set stores the test data itself instead of the feature representations since the embedding space of the feature extractor steadily changes during the test time. The pseudocode for TAST-BN is presented in Appendix B.

## 4 EXPERIMENTS

In this section, we show the effectiveness of our method compared to the state-of-the-art test-time adaptation methods on two standard benchmarks, i.e., domain generalization and image corruption. We compare TAST with the following baseline methods: (1) Pseudo Labeling (PL) (Lee, 2013) fine-tunes the trained classifier using confident pseudo labels based on classifier predictions; (2) PLClf is a modified version of PL that fine-tunes only the last linear classifier; (3) Tent (Wang et al., 2021a)

---

[2]Detailed explanation about the adaptation modules is described in Section 4.1.1. and Appendix A.

[3]We set $\tau$ manually to 0.1 inspired by Oreshkin et al. (2018) for experiments throughout this paper. More experimental results with different $\tau$ are summarized in Appendix C

Table 1: Average accuracy (%) using classifiers learned by ERM on the domain generalization benchmarks. We use ResNet-18 and ResNet-50 as backbone networks. **Bold** indicates the best performance for each benchmark. Underline indicates the best performance among the baseline methods for each benchmark. Most of the baseline methods degrade the classification performance of the trained classifiers on the benchmarks. However, our method consistently outperforms all the baselines on all of the benchmarks.

| Method | Memory usage | Backbone | VLCS | PACS | OfficeHome | TerraIncognita | Avg |
|---|---|---|---|---|---|---|---|
| ERM | | | 74.88±0.46 | 79.29±0.77 | 62.10±0.31 | 40.62±1.19 | 64.22 |
| +Tent | | | 72.88±0.82 | 83.89±0.54 | 60.86±0.39 | 33.70±1.09 | 62.83 |
| +TentAdapter | | | 67.02±1.16 | 80.75±1.01 | 62.64±0.38 | 39.91±0.76 | 62.58 |
| +TentClf | | | 72.96±1.48 | 78.57±1.78 | 59.33±0.62 | 38.30±3.44 | 62.29 |
| +SHOT | | | 65.24±2.29 | 82.36±0.63 | 62.58±0.39 | 33.57±1.04 | 60.94 |
| +SHOTIM | | ResNet-18 | 64.86±2.22 | 82.33±0.61 | 62.57±0.39 | 33.35±1.23 | 60.78 |
| +PL | | | 62.97±2.72 | 70.98±1.78 | 58.20±3.21 | 37.44±7.20 | 57.40 |
| +PLClf | | | 74.89±0.61 | 78.11±2.30 | 61.92±0.41 | 41.78±1.94 | 64.18 |
| +T3A | ✓ | | 77.26±1.49 | 80.83±0.67 | 63.21±0.50 | 40.20±0.60 | 65.38 |
| +TAST (Ours) | ✓ | | **77.27±0.67** | 81.94±0.44 | **63.70±0.52** | **42.64±0.72** | **66.39** |
| +TAST-BN (Ours) | ✓ | | 75.21±2.36 | **87.07±0.53** | 62.79±0.41 | 39.43±2.24 | 66.13 |
| ERM | | | 76.71±0.50 | 83.21±1.14 | 67.13±0.99 | 45.93±1.34 | 68.25 |
| +Tent | | | 72.96±1.27 | 85.16±0.62 | 66.29±0.77 | 37.08±2.04 | 65.37 |
| +TentAdapter | | | 69.65±1.17 | 83.69±1.16 | 67.91±0.89 | 43.89±1.25 | 66.29 |
| +TentClf | | | 75.80±0.68 | 82.66±1.59 | 66.79±0.98 | 43.64±2.59 | 67.22 |
| +SHOT | | | 67.07±0.90 | 84.07±1.23 | 67.65±0.72 | 35.20±0.82 | 63.50 |
| +SHOTIM | | ResNet-50 | 66.93±0.84 | 84.14±1.25 | 67.65±0.77 | 34.37±1.07 | 63.27 |
| +PL | | | 69.41±3.12 | 81.72±4.61 | 62.85±3.05 | 38.09±2.35 | 63.02 |
| +PLClf | | | 75.65±0.88 | 83.33±1.59 | 67.01±1.00 | 46.66±2.12 | 68.16 |
| +T3A | ✓ | | 77.29±0.39 | 83.92±1.13 | 68.26±0.84 | 45.61±1.10 | 68.77 |
| +TAST (Ours) | ✓ | | **77.66±0.48** | 84.11±1.22 | 68.63±0.70 | **47.43±2.09** | **69.46** |
| +TAST-BN (Ours) | ✓ | | 73.52±1.37 | **89.16±0.47** | **68.88±0.50** | 41.47±2.88 | 68.26 |

fine-tunes only the parameters of the BN layers to minimize the prediction entropy of test data; (4) TentAdapter is a modified version of Tent that adds a BN layer between the feature extractor and the last linear classifier, and fine-tunes only the added BN layer; (5) TentClf is a modified version of Tent that fine-tunes only the last linear classifier instead of the BN layers; (6) SHOTIM (Liang et al., 2020) updates the feature extractor to maximize the mutual information between an input and its prediction; (7) SHOT is a method that adds a pseudo-label loss to SHOTIM; (8) T3A predicts the label of the test data by comparing distances between test data and the generated pseudo-prototypes. Originally, SHOT is one of source-free domain adaptation methods which focus on the offline setting, but we compare our method with the online version of SHOT for a fair comparison.

## 4.1 DOMAIN GENERALIZATION

The domain generalization benchmarks are designed to evaluate the generalization ability of the trained classifiers to the unseen domain. The evaluation is performed by a leave-one-domain-out procedure, which uses a domain as a test domain and the remaining domains as training domains. We use the publicly released code [4] of T3A for the domain generalization benchmarks.

### 4.1.1 EXPERIMENTAL SETUP

**Training setup** We test TAST on four domain generalization benchmarks, specifically VLCS (Fang et al., 2013), PACS (Li et al., 2017), OfficeHome (Venkateswara et al., 2017), and TerraIncognita (Beery et al., 2018). For a fair comparison, we follow the training setup including dataset splits and hyperparameter selection method used in T3A. We use residual networks (He et al., 2016) including batch normalization layers with 18 and 50 layers (hereinafter referred to as ResNet-18 and ResNet-50, respectively), which are widely used for classification tasks. We train the networks with various learning algorithms such as ERM and CORAL (Sun and Saenko, 2016). Details about the learning algorithms are explained in Appendix A. The backbone networks are trained with the

---

[4] https://github.com/matsuolab/T3A

Table 2: Ablation studies to evaluate the effects of the number of adaptation module and the nearest neighbor information. We use ResNet-18 trained by ERM. TAST-N is a method that removes adaptation modules from TAST.

| Method | $N_e$ | VLCS | PACS | OfficeHome | TerraIncognita | Avg |
|---|---|---|---|---|---|---|
| ERM | - | 74.88±0.46 | 79.29±0.77 | 62.10±0.31 | 40.62±1.19 | 64.22 |
| +T3A | - | 77.26±1.49 | 80.83±0.67 | 63.21±0.50 | 40.20±0.60 | 65.38 |
| +TAST-N (Ours) | - | 76.20±1.87 | 81.62±0.52 | 63.54±0.63 | 41.88±1.21 | 65.81 |
| +TAST (Ours) | 1 | 75.20±0.77 | 81.23±0.70 | 62.09±0.64 | 42.59±0.41 | 65.28 |
| | 5 | 76.68±0.77 | 81.81±0.13 | 63.51±0.59 | 42.68±0.80 | 66.17 |
| | 10 | 77.43±0.62 | 81.56±0.85 | 63.39±0.56 | 42.60±0.63 | 66.25 |
| | 20 | 77.27±0.67 | 81.94±0.44 | 63.70±0.52 | 42.64±0.72 | 66.39 |

default hyperparameters introduced in Gulrajani and Lopez-Paz (2021). We use a BatchEnsemble (Wen et al., 2020), which is an efficient ensemble method that reduces the computational cost by weight-sharing, for the adaptation modules of TAST. The output dimension of each adaptation module is set to a quarter of the output dimension of the feature extractor[5], e.g., 128 for ResNet-18. We use Kaiming normalization (He et al., 2015) for initializing the adaptation modules at the beginning of test time. We run experiments using four different random seeds. More details on the benchmarks and the training setups can be found in Appendix A. Moreover, a discussion on computation complexity such as runtime comparison is summarized in Appendix A.

**Hyperparameters** For a fair comparison, the baseline methods use the same hyperparameters as in Iwasawa and Matsuo (2021). TAST uses the same set of possible values for each hyperparameter with baseline methods. TAST involves four hyperparameters: the number of gradient steps per adaptation $T$, the number of support examples per each class $M$, the number of nearby support examples $N_s$, and the number of adaptation modules $N_e$. We define a finite set of possible values for each hyperparameter, $N_s \in \{1, 2, 4, 8\}$, $T \in \{1, 3\}$, and $M \in \{1, 5, 20, 50, 100, -1\}$, where $-1$ means to storing all samples without filtering. $N_e$ is set to 20. We use Adam optimizer with a learning rate of 0.001. More details on the hyperparameters can be found in Appendix A. Moreover, refer to Appendix C for the sensitivity analysis on hyperparameters including the test batch sizes.

### 4.1.2 EXPERIMENTAL RESULTS

In Table 1, we summarize the experimental results of test-time adaptation methods using classifiers trained by ERM. Our method consistently improves the performance of the trained classifiers by 2.17% for ResNet-18 and 1.21% for ResNet-50 on average, respectively. TAST also outperforms the baseline methods including the state-of-the-art test-time adaptation method T3A. Compared to T3A, TAST shows better performance by 1.01% for ResNet-18 and 0.69% for ResNet-50 on average, respectively. Especially, we find that our method significantly improves the performance of the trained classifiers in the TerraIncognita benchmark, which is a challenging benchmark in that the trained classifier shows the lowest prediction accuracy. We observe that the performance of the baseline methods, which fine-tune the feature extractors, is lower than that of the classifiers without adaptation, whereas TAST-BN improves the performance of the trained classifiers. Refer to Appendix C for the experimental results of test-time adaptation methods using classifiers trained by different learning algorithms such as CORAL (Sun and Saenko, 2016) and MMD (Li et al., 2018).

**Effect of nearest neighbor information** To understand the effect of nearest neighbor information, we compare Tent and TAST-BN, both of which fine-tine the BN layers. To adjust the BN layers, Tent uses entropy minimization loss, whereas TAST-BN uses the pseudo-label loss using the nearest neighbor information. As shown in Table 1, the performances of TAST-BN is better than those of Tent by 3.3% for ResNet-18 and 2.89% for ResNet-50, respectively. In addition, we consider an ablated variant of TAST, named TAST-N, that removes adaptation modules from TAST. TAST-N is optimization-free and has the same support set configuration as T3A. T3A uses the prototype-based prediction of the test data itself, whereas TAST-N uses the aggregated predicted class distribution

---

[5]More experimental results with different output dimensions are summarized in Appendix C.

of the nearby support examples. As shown in Table 2, the prediction using the nearest neighbor information leads to a performance gain of $0.43\%$ on average.

**Effect of adaptation modules**  TAST adds randomly initialized adaptation modules on top of the trained feature extractor as illustrated in Figure 1 and trains the adaptation modules during test time. For each test batch, we update the adaptation modules $T$ times using pseudo label distributions considering nearest neighbor information. We set $T$ to 1 or 3 throughout all experiments. To verify that the few step updates are sufficient to train the adaptation modules, we conduct experiments with different $T \in \{0, 1, 2, 4, 8\}$. We test on PACS using classifiers learned by ERM while $M$ and $N_s$ are set to $-1$ and one of $\{1, 2, 4, 8\}$. We summarize the experimental results in Figure 2. We observe that the performance of the adapted classifier is better than that of the non-adapted classifier (i.e., $T = 0$) and robust to changes in $T$. Hence, we conjecture that we can obtain a sufficiently good adaptation module with a few-step updates similar to Lee and Chung (2021a).

In addition, to investigate the effect of adaptation modules, we test TAST with a varying number of adaptation modules, e.g., $N_e \in \{1, 5, 10, 20\}$. In Table 2, we find that utilizing a single adaptation module leads to degraded performance than TAST-N. However, TAST with multiple adaptation modules shows improvement over TAST-N and T3A on average.

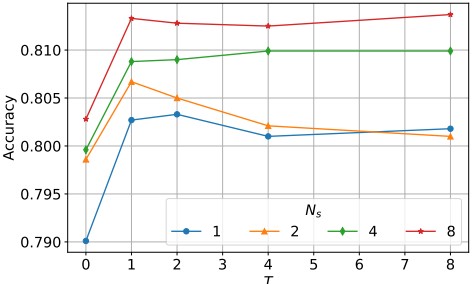

| Method | CIFAR-10C | CIFAR-100C |
|---|---|---|
| No adaptation | 29.14 | 60.35 |
| +SHOT | 15.32 | 41.54 |
| +Tent | 13.95 | 39.04 |
| +PL | 22.34 | 40.06 |
| +T3A | 26.68 | 58.28 |
| +TAST (Ours) | 26.61 | 60.74 |
| +TAST-BN (Ours) | **13.08** | **37.82** |
| +TTT++ | 14.33 | 42.38 |

Figure 2: Sensitivity analysis about $N_s$, the number of nearby support examples, and $T$, the number of gradient steps per adaptation. Average accuracy on test environment A using classifiers learned by ERM on PACS when $M$ is set to $-1$.

Table 3: Average error rate ($\%$) on CIFAR-10C/100C. We test on the highest level of image corruption. **Bold** indicates the best performance for each image corruption.

## 4.2  Image Corruption

The image corruption benchmark is designed to evaluate the robustness of a classifier to unseen corrupted samples when the classifier is trained using clean samples. We use the publicly released code [6] of TTT++ (Liu et al., 2021) for the image corruption benchmark. For a fair comparison, we compare our method with the online version of TTT++, which fine-tunes the feature extractor using the instance discrimination task along with matching the feature statistics of training and test time.

### 4.2.1  Experimental setup

We test the robustness of TAST to image corruption on CIFAR-10/100 (Krizhevsky and Hinton, 2009), which is composed of generic images consisting of 10/100 classes, respectively. To make a corrupted test dataset, we apply 15 types of common image corruptions (e.g., Gaussian noise, shot noise) to the test dataset. We call the corrupted dataset CIFAR-10C/100C (Hendrycks and Dietterich, 2019). We use the highest level (i.e., level-5) of image corruption for this experiment. We use ResNet-50 as a backbone network. For a fair comparison, we use the released trained model of Liu et al. (2021) and the same hyperparameters whenever possible. The number of nearby support examples $N_s$ is set to 1, the number of gradient steps per adaptation $T$ is set to 1, the number of adaptation modules $N_e$ is set to 20, the number of support examples per each class $M$ is set to 100, and the test batch size is set to 128. More experimental results with other hyperparameter combinations are summarized in Appendix C.

---

[6] https://github.com/vita-epfl/ttt-plus-plus

### 4.2.2 EXPERIMENTAL RESULTS

The overall experimental results on CIFAR-10C/100C are summarized in Table 3. We note that the best TTA method which achieves effective adaptation in the image corruption benchmarks can be different from that of the domain generalization benchmarks, since the two benchmarks deal with very different types of domain/distribution shifts. From Table 1 and 3, we can observe that the test-time adaptation algorithms using the frozen feature extractor such as T3A and TAST show poor performance for image corruption benchmarks but better performance for domain generalization benchmarks, compared to those using the adapted feature extractor such as Tent and TAST-BN. Specifically, TAST-BN outperforms all the TTA methods and TTT++, and it achieves performance gains of $1.25\%$ for CIFAR-10C and $4.56\%$ for CIFAR-100C on average, compared to Tent, respectively. Refer to Appendix E for the detailed experimental results on 15 types of image corruptions.

## 5 RELATED WORKS

**Test-time training methods** Test-time training methods fine-tune trained classifiers by the self-supervised learning task used at training time. Sun et al. (2020) uses a rotation prediction task (Feng et al., 2019), which predicts the rotation angle of the rotated images. Liu et al. (2021) use an instance discrimination task (Chen et al., 2020). However, TTA methods, our focus in this paper, have no access to any information related to the training procedure. We empirically demonstrated that our method outperforms the existing test-time training methods on the image corruption benchmark even without the knowledge of the self-supervised learning task.

**Source-free domain adaptation methods** Source-Free Domain Adaptation (SFDA) methods (Liang et al., 2020; Ishii and Sugiyama, 2021; Yeh et al., 2021; Eastwood et al., 2022) aim to adapt trained classifiers to unseen test domains without training dataset. SFDA methods mainly focus on the setting that they can access the whole unlabeled test data, whereas TTA methods can access the online unlabeled test data only. Recently, several SFDA methods using nearest neighbor information (Tang et al., 2021; Yang et al., 2021) have achieved good performances in domain adaptation benchmarks. Especially, NRC (Yang et al., 2021) is built on the similar intuition that a test data and its nearest neighbors share the same label under domain shift. However, unlike NRC, TAST utilizes adaptation module structures and prototype-based classification.

**Ensemble scheme in test-time adaptation** BACS (Zhou and Levine, 2021), which incorporates a Bayesian inference framework into the TTA setting, adapts the trained model to an unseen test domain with a regularization term induced by a posterior approximated at training time. BACS constructs the ensemble of predictive models to obtain diverse labeling for uncertainty estimates at the beginning of training time and trains the models independently during training time. During test time, BACS averages the predictions of the adapted ensemble members. On the other hand, TAST builds an ensemble of adaptation modules to alleviate the issues caused by the random initialization of the modules at the beginning of test time.

## 6 DISCUSSION

We proposed TAST to effectively adapt trained classifiers during test time considering nearest neighbor information. We demonstrated the efficiency and effectiveness of our method by conducting experiments on domain generalization and image corruption benchmarks. To the best of our knowledge, our work is the first one that utilizes an ensemble scheme that is built at test time for test-time adaptation. We expect that adaptation using the ensemble scheme can be combined with the other methods in source-free domain adaptation or test-time training.

One of the limitations of TAST is the extension to large-scale benchmarks. TAST and TAST-BN require good prototypes in the embedding space for prediction and pseudo-labeling. To obtain good prototypes, TAST and TAST-BN construct and update the prototypes using the encountered pseudo-labeled data during the test time. This prototype construction/update, however, can be ineffective for the large-scale benchmarks especially for too many classes and small batch sizes. Detailed discussion of TAST/TAST-BN on large-scale benchmarks and possible improvement of TAST-BN for large-scale benchmarks is described in Appendix D.

ACKNOWLEDGEMENT

This research was supported by the National Research Foundation of Korea under grant 2021R1C1C11008539, and by the Ministry of Science and ICT, Korea, under the IITP (Institute for Information and Communications Technology Panning and Evaluation) grant No.2020-0-00626.

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

## A  BENCHMARK AND IMPLEMENTATION DETAILS

### A.1  DOMAIN GENERALIZATION BENCHMARKS

We test on four domain generalization benchmarks, specifically VLCS (Fang et al., 2013), PACS (Li et al., 2017), OfficeHome (Venkateswara et al., 2017), and TerraIncognita (Beery et al., 2018). VLCS is composed of photographic images from four different datasets (PASCAL VOC207 (Everingham et al., 2010), LableMe (Russell et al., 2008), Caltech 101 (Fei-Fei et al., 2007), and SUN09 (Choi et al., 2010)), consisting of 10,729 examples of 5 categories (bird, car, chair, dog, and person). PACS is composed of images of objects from four different domains (photo, art, cartoon, and sketch), consisting of 9,991 examples of 7 categories (dog, elephant, giraffe, guitar, horse, house, and person). OfficeHome is composed of images of objects in the office and home from 4 different domains (artistic images, clip art, product, and real-world images), consisting of 15,588 examples of 65 categories (e.g., alarm clock, backpack, and batteries). TerraIncognita is composed of wild animal images taken from 4 different locations (L100, L38, L43, and L46), consisting of 24,788 examples of 10 classes.

### A.2  IMPLEMENTATION DETAILS ON DOMAIN GENERALIZATION BENCHMARKS

We follow the dataset splits and the hyperparameter selection method used in T3A. We split each dataset of training domains into training and validation sets. The training and validation sets are used for network training and hyperparameter selection, respectively. Specifically, we split each dataset into $80\%$ and $20\%$ and use the smaller set as the validation set. We choose the hyperparameters that maximize the validation accuracy of the adapted classifier. This hyperparameter selection method is called the training-domain validation. We train backbone networks using four different learning algorithms: ERM, CORAL, MMD, and Mixup. ERM is explained in Section 2 of the manuscript; CORAL aims to obtain domain-invariant representations by aligning covariance matrices of training data and test data; MMD tries to match the training and test data distributions using the MMD measure; Mixup trains classifiers using mixed images/features and mixed labels created by linear interpolation of examples from the training domains. We run experiments using four different random seeds: 0, 1, 2, and 3.

All the hyperparameters for training and test-time adaptation are taken from T3A and DomainBed. We train the network with Adam optimizer with default hyperparameters introduced in DomainBed, e.g., a learning rate of 0.00005, a weight decay of 0, a dropout rate of 0, and a batch size of 32. In addition to the hyperparameters for test-time adaptation described in Section 4.1.1 of the manuscript, there is one more hyperparameter $\beta$ for the baseline methods. The learning rate for test-time adaptation is obtained by multiplying $\beta$ to the learning rate used in training time. We set the confidence threshold for PL and PLClf to 0.9. The possible values for $\beta$ are set to 0.1, 1.0, and 10.0. For TAST-BN, we restrict the size of the whole support set to $150$ due to effective memory usage and reduced runtime since the test data and the support examples are fed into the classifier for every test batch.

### A.3  IMPLEMENTATION DETAILS ON IMAGE CORRUPTION BENCHMARKS

We use the same hyperparameters introduced in TTT++. We train ResNet-50 for 1000 epochs using the classification and instance discrimination tasks jointly. The weight on the instance discrimination task for balancing the two tasks is set to 0.1. For the instance discrimination task, we use the same data augmentation schemes of TTT++, e.g., RandomResizeCrop, RandomHorizontalCrop, HorizontalFlip, ColorJitter, RandomGrayscale, and Normalization. We set the batch size for training the networks to 256. At test time, PL, SHOT, and TTT++ use SGD optimizer with a learning rate of 0.001 and a momentum of 0.9. On the other hand, Tent, TAST, and TAST-BN use Adam optimizer with a learning rate of 0.001. We set the batch size to 128 during the test time due to effective memory usage. We run experiments using four different random seeds: 0, 1, 2, and 3. We set the confidence threshold for PL and PLClf to 0.9. For PL, we adjust only the BN layers in the trained model as in Tent. For TAST-BN, we restrict the size of the whole support set to $200$. However, even in CIFAR-100C experiments, we can store only two support examples per class if the support set size is fixed at 200. Thus, we do not restrict the size of support set for TAST-BN on CIFAR-100C.

## A.4 Runtime comparison

Table 4: Mean runtime (sec) to adapt classifiers that use ResNet-18 as a backbone network with a single hyperparameter combination ($T = 1, N_s = 8, M = -1$).

| Method | VLCS | PACS | OfficeHome | TerraIncognita |
|---|---|---|---|---|
| Tent | 53.32 | 15.17 | 50.56 | 76.48 |
| TentAdapter | 0.59 | 0.43 | 0.64 | 0.92 |
| TentClf | 0.52 | 0.40 | 0.60 | 0.81 |
| SHOT | 55.10 | 20.97 | 54.66 | 77.08 |
| SHOTIM | 54.82 | 20.73 | 54.63 | 77.00 |
| PL | 55.03 | 20.75 | 54.53 | 77.02 |
| PLClf | 0.57 | 0.43 | 0.62 | 0.92 |
| T3A | 0.62 | 0.58 | 3.44 | 1.61 |
| TAST (Ours) | 7.71 | 6.92 | 12.74 | 23.69 |
| TAST-BN (Ours) | 81.54 | 73.93 | 114.33 | 179.48 |

We conduct our experiments on TITAN XP. We report the average runtime spent to adapt classifiers that use ResNet-18 as a backbone network in Table 4. We note that TAST, which updates the support set and the adaptation modules, requires only 1/3 to 1/4 running time compared to the methods that update the entire feature extractors, e.g. SHOT or SHOTIM. On the other hand, TAST-BN, which updates the support set as well as the BN layer, requires more running time (about 2x) compared to SHOT or SHOTIM. The overhead is not significant though due to the online setting.

## A.5 Details about adaptation modules

We use BatchEnsemble (BE) for the adaptation modules of our method. BE is a simple and efficient ensemble method that greatly reduces the computational cost by weight-sharing. Each ensemble member of BE is composed of two layers with a shared weight and rank-one factors. Specifically, the weight matrix of $j$-th ensemble member is $W \circ r_j s_j^T$ where $W$ is a shared weight and $r_j s_j^T$ is the rank-one factor of $j$-th ensemble member. Although the existing deep ensemble (DE) methods do not share any weights, all ensemble members share $W$, and thus BE reduces the number of parameters compared to DE. Moreover, unlike DE, only the last layer of all ensemble members of BE are different, and thus it can be easily vectorized and trained simultaneously. Therefore, BE greatly reduces the computation cost.

The adaptation module structure is used in many fields such as self-supervised learning (which is often called "projection head"). Although the existing methods mainly focus on training time, TAST focuses on test time. For example, SimCLR (Chen et al., 2020) adds a projection head on the top of a feature extractor at the beginning of training time and trains the feature extractor and the projection head with an instance discrimination loss. After the training time, for downstream tasks, SimCLR uses feature extractor outputs rather than projection head ones. However, TAST adds adaptation modules at the beginning of test time and trains the modules with the nearest neighbor-based pseudo-label distribution. To predict the label of test data, we use the averaged predicted class distribution from the adaptation modules.

## B Pseudocode for TAST-BN

We present the pseudocode for TAST-BN in Algorithm 2. TAST-BN fine-tunes the BN layers in the feature extractor instead of adaptation modules. Since the embedding space of the feature extractor steadily changes, the support set stores the test data itself instead of the feature representations. Formally, a support set $\mathbb{S}_t = \{\mathbb{S}_t^1, \mathbb{S}_t^2, \dots, \mathbb{S}_t^K\}$ is a set of test samples until time $t$. The support set is initialized as an empty set. At the time $t$, the support set is updated as

$$\mathbb{S}_t^k = \begin{cases} \mathbb{S}_{t-1}^k \cup \{x_t\}, & \text{if } \arg\max_c p_c = k \\ \mathbb{S}_{t-1}^k, & \text{otherwise,} \end{cases} \tag{9}$$

---

**Algorithm 2** TAST-BN

---

**Require:** Feature extractor $f_\theta$, test batch $\mathbb{B}$, support set $\mathbb{S}$, number of gradient steps per adaptation $T$, number of support examples per each class $M$, number of nearby support examples $N_s$, learning rate $\alpha$
**Ensure:** Predictions $\hat{y}_x$ for all $x \in \mathbb{B}$
  Update the support set $\mathbb{S}$ with eq. (9) in Section B
  Retrieve the nearest neighbors $\mathcal{N}(x; \mathbb{S})$ for all $x \in \mathbb{B}$ with eq. (10) in Section B
  **for** $t = 1 : T$ **do**
    Compute prototypes $\{\mu_k\}_{k=1}^K$ using the support set in the embedding space of $f_\theta$
    **for** $z \in \mathcal{N}(x; \mathbb{S})$ **do**
      $p^{\text{proto}}(k|z) \leftarrow \frac{\exp(-d(f_\theta(z), \mu_k)/\tau)}{\sum_c \exp(-d(f_\theta(z), \mu_c)/\tau)}, k = 1, 2, \ldots, K$
    **end for**
    **for** $x \in \mathbb{B}$ **do**
      $\hat{p}^{\text{TAST}}(k|x) \leftarrow \frac{1}{N_s} \sum_{z \in \mathcal{N}(x; \mathbb{S})} \mathbb{1}[\arg\max_c p^{\text{proto}}(c|z) = k], k = 1, 2, \ldots, K$
      $p^{\text{proto}}(k|x) \leftarrow \frac{\exp(-d(f_\theta(x), \mu_k)/\tau)}{\sum_c \exp(-d(f_\theta(x), \mu_c)/\tau)}, k = 1, 2, \ldots, K$
    **end for**
    $\theta \leftarrow \theta - \alpha \nabla_\theta \frac{1}{|\mathbb{B}|} \sum_{x \in \mathbb{B}} \text{CE}(\hat{p}^{\text{TAST}}(\cdot|x), p^{\text{proto}}(\cdot|x))$
  **end for**
  **for** $x \in \mathbb{B}$ **do**
    $p^{\text{TAST}}(k|x) \leftarrow \frac{1}{N_s} \sum_{z \in \mathcal{N}(x; \mathbb{S})} p^{\text{proto}}(k|z), k = 1, 2, \ldots, K$
    $\hat{y}_x \leftarrow \arg\max_c p^{\text{TAST}}(c|x)$
  **end for**

---

where $p_k$ is the likelihood the classifier assigns $x_t$ to the class $k$. Using the support set, we retrieve $N_s$ nearby support examples of $x$ in the embedding space of $f_\theta$, i.e.,

$$\mathcal{N}(x; \mathbb{S}) := \{z \in \mathbb{S} | d(f_\theta(x), f_\theta(z)) \leq \beta_x\}, \tag{10}$$

where $\beta_x$ is the distance between $x$ and the $N_s$-th nearest neighbor of $x$ from $\mathbb{S}$ in the embedding space of $f_\theta$. Then, we generate a pseudo label distribution for the test data and fine-tune the BN layers to match the nearest neighbor-based pseudo label and a prototype-based class distributions for the test data with the same procedure described in Section 3 of the manuscript.

## C  ADDITIONAL EXPERIMENTS

### C.1  EXPERIMENTAL RESULTS USING CLASSIFIERS TRAINED BY DIFFERENT LEARNING ALGORITHMS

In Table 5, we show the results of test-time adaptation methods using classifiers trained by three different learning algorithms, namely CORAL, MMD, and Mixup. TAST consistently enhances the performance of the trained classifiers on the benchmarks by $1.73\%$, $1.81\%$, and $2.30\%$ on average using the classifiers trained by CORAL, MMD, and Mixup, respectively. We find that TAST has a minor performance gain compared to the results in Table 1 of manuscript, whereas it surpasses T3A on most of benchmarks. Compared to T3A, TAST shows better performance on the benchmarks by $0.21\%$, $0.14\%$, and $0.40\%$ on average with the classifiers trained by CORAL, MMD, and Mixup, respectively. Refer to Section 4 in Appendix E for the experimental results of the other baseline methods.

### C.2  FINE-TUNING BOTH ADAPTATION MODULES AND BN LAYERS SIMULTANEOUSLY

We consider a method, named TAST-both, that fine-tunes both the attached adaptation modules and the BN layers in the feature extractor simultaneously. Table 6 reports the experimental results using classifiers learned by ERM on domain generalization benchmarks. We use ResNet-18 as a backbone network. As shown in Table 6, TAST-both shows worse performance than TAST-BN and TAST. We conjecture that the random initialization of adaptation modules and the changes in feature representation due to BN layer training negatively affect the learning of the other layers.

Table 5: Average accuracy (%) on domain generalization benchmarks using classifiers trained by different learning algorithms, namely CORAL, MMD, and Mixup. We use ResNet-18 as a backbone network. **Bold** indicates the best performance for each benchmark. TAST and TAST-BN consistently improve the performance of the trained classifiers and they outperform T3A on most of the benchmarks.

| Method | VLCS | PACS | OfficeHome | TerraIncognita | Avg |
|---|---|---|---|---|---|
| CORAL | 74.00±1.13 | 81.00±0.79 | 62.78±0.06 | 36.51±2.35 | 63.57 |
| +T3A | 75.49±1.67 | 82.75±0.51 | 63.72±0.32 | 38.39±1.39 | 65.09 |
| +TAST (Ours) | 74.82±2.43 | 83.16±0.81 | **64.00±0.25** | **39.21±1.75** | 65.30 |
| +TAST-BN (Ours) | **77.01±0.36** | **87.21±0.57** | 62.98±0.23 | 37.45±1.11 | **66.16** |
| MMD | 74.90±0.50 | 81.06±0.92 | 62.20±0.48 | 35.73±2.70 | 63.47 |
| +T3A | **77.28±0.45** | 82.52±0.53 | 63.34±0.55 | 37.40±1.86 | 65.14 |
| +TAST (Ours) | 76.21±0.79 | 83.29±0.26 | **63.49±0.49** | 38.12±2.47 | 65.28 |
| +TAST-BN (Ours) | 76.06±0.89 | **86.35±0.76** | 63.22±0.26 | **39.46±1.63** | **66.27** |
| Mixup | 74.97±0.86 | 78.29±0.88 | 61.83±0.88 | 41.04±1.01 | 64.03 |
| +T3A | **78.43±0.76** | 81.91±0.54 | 63.49±0.86 | 39.89±0.90 | 65.93 |
| +TAST (Ours) | 77.19±0.80 | 82.85±0.36 | **63.83±0.74** | 41.44±1.67 | 66.33 |
| +TAST-BN (Ours) | 76.89±0.86 | **87.14±0.56** | 62.09±0.86 | **42.70±1.90** | **67.21** |

Table 6: Average accuracy (%) using classifiers trained by ERM on domain generalization benchmarks. We use ResNet-18 as a backbone network. TAST-both is a method fine-tunes both the attached adaptation modules and the BN layers simultaneously. TAST-both shows worse performances than TAST-BN and TAST.

| Method | $N_e$ | VLCS | PACS | OfficeHome | TerraIncognita | avg |
|---|---|---|---|---|---|---|
| ERM | - | 74.88±0.46 | 79.30±0.77 | 62.09±0.31 | 40.63±1.19 | 64.22 |
| +T3A | - | 77.26±1.49 | 80.83±0.67 | 63.21±0.50 | 40.20±0.60 | 65.38 |
| +TAST-N | - | 76.20±1.87 | 81.62±0.52 | 63.54±0.63 | 41.88±1.21 | 65.81 |
| +TAST-BN | - | 75.21±2.36 | 87.07±0.53 | 62.79±0.41 | 39.43±2.24 | 66.13 |
| +TAST | 1 | 75.20±0.77 | 81.23±0.70 | 62.09±0.64 | 42.59±0.41 | 65.28 |
| | 5 | 76.68±0.77 | 81.81±0.13 | 63.51±0.59 | 42.68±0.80 | 66.17 |
| | 10 | 77.43±0.62 | 81.56±0.85 | 63.39±0.56 | 42.60±0.63 | 66.25 |
| | 20 | 77.27±0.67 | 81.94±0.44 | 63.70±0.52 | 42.64±0.72 | 66.39 |
| +TAST-both | 1 | 73.35±0.57 | 84.85±0.56 | 61.70±0.39 | 39.27±2.05 | 64.79 |
| | 5 | 73.88±0.35 | 84.99±0.63 | 61.81±0.44 | 39.16±1.57 | 64.96 |
| | 10 | 73.66±1.57 | 85.13±0.27 | 62.03±0.52 | 38.50±1.25 | 64.83 |
| | 20 | 75.16±0.17 | 85.52±0.05 | 62.01±0.67 | 38.54±1.52 | 65.31 |

## C.3 EXPERIMENTAL RESULTS USING DIFFERENT HYPERPARAMETERS ON CIFAR-10C

In Table 4 of the manuscript, we report the experimental results when $N_s$ and $M$ are set to 1 and 100 on the CIFAR-10C, respectively. In Table 7, we summarize the experimental results using different combinations of $N_s$ and $M$ on the CIFAR-10C. There are two observations in Table 7: (1) T3A has shown the best performances when $M$ is set to 100; and (2) TAST and TAST-BN perform better with smaller $N_s$.

## C.4 SENSITIVITY ANALYSIS ON HYPERPARAMETERS

We follow the hyperparameter selection method used in T3A. We split the dataset of training domains into training and validation sets. The validation set is used to select hyperparameters that maximize the validation accuracy of the adapted classifier. On the other hand, for the image corruption benchmark, we use manually determined hyperparameters as in Tent. Thus, we summarized experimental results on other combinations of hyperparameters in Table 8-11.

Additionally, we investigate the sensitivity of two hyperparameters which are set manually throughout all experiments, the softmax temperature $\tau$ and the output dimension of adaptation modules

Table 7: Average error rate (%) in the online setting on CIFAR-10C with different hyperparameters.

| Method | $M$ | $N_s$ | gauss | brit | contr | defoc | elast | fog | frost | glass | impul | jpeg | motn | pixel | shot | snow | zoom |
|---|---|---|---|---|---|---|---|---|---|---|---|---|---|---|---|---|---|
| No adaptation | - | - | 48.73 | 7.01 | 13.27 | 11.84 | 23.38 | 29.41 | 28.24 | 50.78 | 57.00 | 19.46 | 23.38 | 47.88 | 44.00 | 21.93 | 10.84 |
| +T3A | 1 | - | 44.56 | 8.28 | 13.27 | 13.45 | 22.18 | 28.59 | 27.18 | 46.43 | 55.11 | 18.96 | 22.59 | 42.92 | 40.32 | 21.77 | 10.53 |
| +T3A | 5 | - | 44.65 | 7.94 | 14.11 | 13.34 | 22.67 | 29.00 | 28.57 | 45.92 | 56.03 | 19.67 | 24.16 | 40.18 | 40.55 | 22.61 | 12.02 |
| +T3A | 20 | - | 44.26 | 7.72 | 13.82 | 13.35 | 22.24 | 28.71 | 28.36 | 45.49 | 55.87 | 19.34 | 23.76 | 39.82 | 40.38 | 22.44 | 12.20 |
| +T3A | 50 | - | 42.82 | 7.43 | 13.65 | 12.36 | 22.15 | 28.54 | 27.69 | 44.42 | 54.91 | 19.13 | 22.83 | 38.33 | 38.53 | 22.09 | 11.15 |
| +T3A | 100 | - | 41.87 | 7.30 | 13.61 | 11.99 | 22.06 | 28.52 | 27.13 | 44.10 | 54.26 | 18.71 | 22.54 | 37.53 | 37.84 | 21.97 | 10.72 |
| +T3A | -1 | - | 43.83 | 7.33 | 13.56 | 11.63 | 22.11 | 29.07 | 27.56 | 46.79 | 55.16 | 18.73 | 22.77 | 41.16 | 39.58 | 22.23 | 10.34 |
| +TAST | 1 | 1 | 47.24 | 8.68 | 12.93 | 16.74 | 22.31 | 28.66 | 27.23 | 48.76 | 55.97 | 19.21 | 22.63 | 48.32 | 42.80 | 21.57 | 10.34 |
| +TAST | 5 | 1 | 47.19 | 9.78 | 15.88 | 15.58 | 24.30 | 30.94 | 29.70 | 48.66 | 58.05 | 21.49 | 27.57 | 41.00 | 44.21 | 24.58 | 14.75 |
| +TAST | 5 | 2 | 48.08 | 11.34 | 17.73 | 17.10 | 25.93 | 31.99 | 30.54 | 49.77 | 58.54 | 23.72 | 29.55 | 44.09 | 45.55 | 26.23 | 16.41 |
| +TAST | 5 | 4 | 48.53 | 10.95 | 17.25 | 16.82 | 25.58 | 31.78 | 30.19 | 50.00 | 58.86 | 23.67 | 29.35 | 43.37 | 45.35 | 25.63 | 16.26 |
| +TAST | 20 | 1 | 43.58 | 7.74 | 14.01 | 12.97 | 21.90 | 28.73 | 27.76 | 45.62 | 55.20 | 19.78 | 23.63 | 38.77 | 39.22 | 22.73 | 12.13 |
| +TAST | 20 | 2 | 44.38 | 8.22 | 14.51 | 13.73 | 22.28 | 29.07 | 28.66 | 46.30 | 55.92 | 20.49 | 24.39 | 40.60 | 40.38 | 23.37 | 12.76 |
| +TAST | 20 | 4 | 44.41 | 8.19 | 14.26 | 13.60 | 22.22 | 29.05 | 28.81 | 46.17 | 56.05 | 20.09 | 24.24 | 40.33 | 40.39 | 23.18 | 12.67 |
| +TAST | 50 | 1 | 42.47 | 7.37 | 13.65 | 12.14 | 21.48 | 28.30 | 26.88 | 44.99 | 54.52 | 19.26 | 22.76 | 37.59 | 37.73 | 22.06 | 11.03 |
| +TAST | 50 | 2 | 42.79 | 7.62 | 14.11 | 12.30 | 21.57 | 28.73 | 27.50 | 45.26 | 55.18 | 19.96 | 23.10 | 38.99 | 38.43 | 22.43 | 11.25 |
| +TAST | 50 | 4 | 42.89 | 7.54 | 13.90 | 12.15 | 21.45 | 28.51 | 27.48 | 45.10 | 54.93 | 19.71 | 22.87 | 38.94 | 38.21 | 22.29 | 11.02 |
| +TAST | 100 | 1 | 42.02 | 7.34 | 13.55 | 11.86 | 21.38 | 28.58 | 26.51 | 44.99 | 54.19 | 18.96 | 22.55 | 37.08 | 37.62 | 21.84 | 10.64 |
| +TAST | 100 | 2 | 42.35 | 7.61 | 13.89 | 11.95 | 21.50 | 28.75 | 27.06 | 45.15 | 54.55 | 19.58 | 22.77 | 38.09 | 37.64 | 21.99 | 10.73 |
| +TAST | 100 | 4 | 42.34 | 7.50 | 13.80 | 11.72 | 21.45 | 28.32 | 26.89 | 44.75 | 54.46 | 19.18 | 22.54 | 38.00 | 37.61 | 21.97 | 10.67 |
| +TAST | -1 | 1 | 45.20 | 7.44 | 14.05 | 11.55 | 22.87 | 30.19 | 27.87 | 50.28 | 57.07 | 19.65 | 22.95 | 41.99 | 41.35 | 22.65 | 10.20 |
| +TAST | -1 | 2 | 44.86 | 7.45 | 13.88 | 11.62 | 22.03 | 29.38 | 28.03 | 50.37 | 58.25 | 20.02 | 22.73 | 42.67 | 41.04 | 22.48 | 10.30 |
| +TAST | -1 | 4 | 44.93 | 7.36 | 13.64 | 11.17 | 21.57 | 28.82 | 28.17 | 49.63 | 58.74 | 19.64 | 22.41 | 43.38 | 41.02 | 22.15 | 9.86 |
| +TAST-BN | 1 | 1 | 19.46 | 11.95 | 11.02 | 12.84 | 21.54 | 19.38 | 16.64 | 26.03 | 27.76 | 17.93 | 15.29 | 14.01 | 19 | 17.94 | 11.78 |
| +TAST-BN | 5 | 1 | 16.21 | 8.4 | 8.48 | 9.56 | 17.92 | 15.59 | 13.06 | 23.1 | 23.68 | 13.58 | 12.79 | 11.06 | 14.85 | 14.02 | 8.36 |
| +TAST-BN | 5 | 2 | 17.26 | 9.27 | 9.23 | 10.51 | 19.51 | 16.17 | 13.86 | 24.32 | 24.69 | 14.57 | 13.81 | 12.05 | 15.98 | 15.06 | 8.95 |
| +TAST-BN | 5 | 4 | 18.89 | 10.29 | 11.05 | 13.23 | 20.66 | 16.99 | 14.79 | 24.95 | 26.1 | 16.12 | 17.02 | 13.57 | 18.42 | 17.11 | 10.22 |
| +TAST-BN | 20 | 1 | 14.91 | 7.68 | 7.81 | 8.62 | 16.81 | 15.10 | 12.25 | 21.82 | 22.54 | 12.38 | 11.67 | 10.34 | 13.77 | 12.99 | 7.57 |
| +TAST-BN | 20 | 2 | 15.11 | 7.85 | 7.96 | 8.75 | 17.00 | 15.02 | 12.32 | 22.07 | 22.54 | 12.57 | 11.99 | 10.50 | 13.98 | 13.12 | 7.70 |
| +TAST-BN | 20 | 4 | 15.00 | 7.98 | 8.00 | 8.71 | 16.87 | 14.89 | 12.24 | 21.88 | 22.43 | 12.55 | 11.89 | 10.47 | 14.06 | 13.06 | 7.59 |

$d_\phi$. We set $\tau$ and $d_\phi$ to 0.1 and $d_z/4$, where $d_z$ is the output dimension of the feature extractor. In Table 8-11, we report the average accuracy of the adapted classifier by TAST with the different combinations of $\tau$ and $d_\phi$. In the experiments, we use ResNet-18 as a backbone network trained by ERM on PACS, which is one of the domain generalization benchmarks. We experimentally show that the performance of TAST is robust to changes in $\tau$ and $d_\phi$. We especially think that the classification performance of TAST is not significantly affected by changes in $\tau$ because $\tau$ affects both the prototype-based predicted class distribution of test data and the new pseudo-label distribution using nearest neighbor information and then we train the adaptation modules with the cross-entropy loss affected by $\tau$ only a few times per each test batch during test time. Moreover, we can observe a similar classification performance regardless of the dimension of adaptation modules similar to Chen et al. (2020).

Table 8: Sensitivity analysis about the softmax temperature $\tau$ and the output dimension of adaptation modules $d_\phi$. Average accuracy on test environment A using classifiers learned by ERM on PACS.

| testenv:A | | $d_z$ | $d_z/2$ | $d_z/4$ (used) | $d_z/8$ | $d_z/16$ |
|---|---|---|---|---|---|---|
| | 10 | 0.8025 | 0.8028 | 0.8024 | 0.8023 | 0.8026 |
| | 1 | 0.8034 | 0.8038 | 0.8029 | 0.8034 | 0.8034 |
| $\tau$ | 0.1 (used) | 0.8031 | 0.8038 | 0.8056 | 0.8038 | 0.8034 |
| | 0.01 | 0.8026 | 0.8018 | 0.8025 | 0.8028 | 0.8023 |
| | 0.001 | 0.8020 | 0.8030 | 0.8023 | 0.8019 | 0.8001 |

We used the test batch size as in T3A and Tent for domain generalization and image corruption benchmarks, respectively, as described in Appendix A and Section 4 of the manuscript. We summarize experimental results using different test batch size. We conduct experiments using classifiers, which have ResNet-18 backbone networks, learned by ERM on PACS. As shown in Table 12, we can find that Tent and PL show reduced performance in experiments using smaller test batch size, but T3A, TAST, and TAST-BN are robust to changes in test batch size.

Table 9: Sensitivity analysis about the softmax temperature $\tau$ and the output dimension of adaptation modules $d_\phi$. Average accuracy on test environment C using classifiers learned by ERM on PACS.

| testenv:C | | $d_z$ | $d_z/2$ | $d_\phi$
$d_z/4$ (used) | $d_z/8$ | $d_z/16$ |
|---|---|---|---|---|---|---|
| | 10 | 0.7842 | 0.7838 | 0.7841 | 0.7838 | 0.7835 |
| | 1 | 0.7830 | 0.7836 | 0.7829 | 0.7837 | 0.7836 |
| $\tau$ | 0.1 (used) | 0.7817 | 0.7815 | 0.7826 | 0.7816 | 0.7824 |
| | 0.01 | 0.7810 | 0.7816 | 0.7842 | 0.7832 | 0.7825 |
| | 0.001 | 0.7811 | 0.7796 | 0.7802 | 0.7806 | 0.7803 |

Table 10: Sensitivity analysis about the softmax temperature $\tau$ and the output dimension of adaptation modules $d_\phi$. Average accuracy on test environment P using classifiers learned by ERM on PACS.

| testenv:P | | $d_z$ | $d_z/2$ | $d_\phi$
$d_z/4$ (used) | $d_z/8$ | $d_z/16$ |
|---|---|---|---|---|---|---|
| | 10 | 0.9611 | 0.9611 | 0.9614 | 0.9613 | 0.9609 |
| | 1 | 0.9614 | 0.9614 | 0.9611 | 0.9614 | 0.9616 |
| $\tau$ | 0.1 (used) | 0.9615 | 0.9620 | 0.9644 | 0.9613 | 0.9606 |
| | 0.01 | 0.9621 | 0.9609 | 0.9613 | 0.9606 | 0.9604 |
| | 0.001 | 0.9611 | 0.9615 | 0.9607 | 0.9606 | 0.9613 |

Table 11: Sensitivity analysis about the softmax temperature $\tau$ and the output dimension of adaptation modules $d_\phi$. Average accuracy on test environment S using classifiers learned by ERM on PACS.

| testenv:S | | $d_z$ | $d_z/2$ | $d_\phi$
$d_z/4$ (used) | $d_z/8$ | $d_z/16$ |
|---|---|---|---|---|---|---|
| | 10 | 0.7180 | 0.7208 | 0.7186 | 0.7155 | 0.7185 |
| | 1 | 0.7175 | 0.7214 | 0.7180 | 0.7211 | 0.7202 |
| $\tau$ | 0.1 (used) | 0.7211 | 0.7222 | 0.7252 | 0.7216 | 0.7220 |
| | 0.01 | 0.7236 | 0.7232 | 0.7223 | 0.7225 | 0.7209 |
| | 0.001 | 0.7246 | 0.7196 | 0.7251 | 0.7235 | 0.7225 |

Table 12: Ablation studies to evaluate the effects of the test batch size.

| Methods | Batch size $B$
8 | 16 | 32 (used) | 64 | 128 |
|---|---|---|---|---|---|
| ERM | 79.31± 0.75 | 79.30±0.76 | 79.29±0.77 | 79.29±0.76 | 79.28±0.70 |
| +Tent | 77.52±0.49 | 81.16±0.46 | 83.89±0.54 | 83.90±0.54 | 83.85±0.12 |
| +SHOT | 81.44±0.32 | 82.12±0.75 | 82.36±0.63 | 83.18±0.34 | 82.95±0.33 |
| +PL | 67.90±4.26 | 70.33±3.53 | 70.98±1.78 | 77.52±2.89 | 78.90±0.39 |
| +T3A | 81.21±0.76 | 81.22±0.69 | 80.83±0.67 | 81.20±0.73 | 81.27±0.6 |
| +TAST (ours) | 81.81±0.35 | 81.52±1.04 | 81.94±0.44 | 81.92±0.87 | 81.69±0.64 |
| +TAST-BN (ours) | **86.78±0.78** | **86.66±1.24** | **87.07±0.53** | **86.90±0.49** | **86.92±0.42** |

## D  TAST ON IMAGENET-C

ImageNet-C is an image corruption benchmark such as CIFAR-10/100C, but it is a large-scale benchmark composed of larger images from more diverse classes. ImageNet-C is challenging for the existing test-time adaptation/training methods including TTT++. Also TAST and TAST-BN may struggle with ImageNet-C, since TAST and TAST-BN require prototypes to represent each class in the embedding space. To obtain good prototypes, a sufficient amount of data per class is required, but we have no access to any labeled data due to TTA settings. Pseudo-labeling alleviates this issue on CIFAR-10/100C, but not on ImageNet-C due to the following concerns:

- The prototype updates of TAST and TAST-BN are based on the estimated labels of test data by the classifier, not the ground-truth labels. Under test-time domain shift, classifier bias may occur, which may result in assigning most test data only to a subset of classes. As observed in Chen et al. (2022), the classifier bias often occurs under the covariate shift such as image corruption and style transfer. Then, even after a large number of batch updates which cover all the ground-truth classes by at least one sample, some prototypes may have not been updated since no previous test data has been classified to those classes. For example, we found that for the experiments with Gaussian noise, it took 768 batches out of 782 batches until all the prototypes were updated at least by once.

- Since the number of classes (1000) is much larger than the test batch size (64), few prototypes for our method are updated per each test batch while the remaining prototypes remain unupdated. It might affect the performance of the prototype-based classification. To address this issue, it might require a batch size larger than 1000, which is impossible due to the hardware cost.

When the number of classes (1000) is much larger than the test batch size (64), obtaining good prototypes for TAST-BN can be difficult especially at the early stage of test time as explained above. To alleviate the concerns, we consider a variant of TAST-BN, in which the prototypes are initialized with the weight of the last linear classifier as in TAST and fixed during the test time. We call this variant TAST-BN (w/ fixed prototypes). In Table 13, we report the experimental results (test accuracy) on ImageNet-C with severity level 5 when we set $(N_s, M, T)$ to $(1, -1, 1)$.

Table 13: Accuracy of TAST-BN (w/ fixed prototypes) on ImageNet-C

| method | brit | contr | defoc | elast | fog | frost | gauss | glass | impul | jpeg | motn | pixel | shot | snow | zoom | avg |
|---|---|---|---|---|---|---|---|---|---|---|---|---|---|---|---|---|
| NoAdapt | 0.5893 | 0.0543 | 0.1792 | 0.1695 | 0.2442 | 0.2331 | 0.0221 | 0.0982 | 0.0185 | 0.3165 | 0.1478 | 0.2061 | 0.0293 | 0.1689 | 0.2250 | 0.1801 |
| TAST-BN (w/ fixed prototypes) | 0.6498 | 0.1926 | 0.1670 | 0.4495 | 0.4960 | 0.3422 | 0.1665 | 0.1645 | 0.1742 | 0.4183 | 0.2826 | 0.5040 | 0.1728 | 0.3615 | 0.4014 | 0.3295 |

Of course, one can still update the prototypes over the test time, but the performance gain from the updating may not be as significant as before. Nonetheless, from the result of Table 13, we can see that the effective adaptation on ImageNet-C can be achieved with the combination of the prototype approach and self-training (entropy minimization) method of TAST-BN (w/ fixed prototypes).

## E  FULL RESULTS

Table 14: Full results using classifiers trained by ERM for Table 1 of the manuscript on VLCS. We use ResNet-18 as a backbone network.

| Method | C | L | S | V | Avg |
|---|---|---|---|---|---|
| ERM | 94.70±1.33 | 63.79±1.30 | 67.90±1.97 | 73.15±1.37 | 74.88 |
| +Tent | 89.82±2.89 | 61.98±1.10 | 65.51±1.91 | 74.21±1.61 | 72.88 |
| +TentAdapter | 79.80±4.74 | 58.51±1.44 | 61.62±0.92 | 68.14±1.74 | 67.02 |
| +TentClf | 94.75±1.43 | 63.74±1.41 | 67.92±2.22 | 65.40±6.91 | 72.96 |
| +SHOT | 91.45±6.83 | 48.26±1.77 | 54.75±2.59 | 66.51±1.25 | 65.24 |
| +SHOTIM | 90.28±7.00 | 47.96±1.45 | 54.66±2.47 | 66.52±1.19 | 64.86 |
| +PL | 93.57±2.24 | 53.82±2.51 | 50.58±9.50 | 53.91±2.78 | 62.97 |
| +PLClf | 94.67±1.38 | 63.64±1.31 | 67.90±2.21 | 73.34±1.00 | 74.89 |
| +T3A | 97.52±1.99 | 65.32±2.24 | 70.70±3.48 | 75.51±1.75 | 77.26 |
| +TAST (Ours) | 99.17±0.60 | 65.87±1.90 | 68.13±1.76 | 75.92±1.75 | 77.27 |
| +TAST-BN (Ours) | 92.60±8.66 | 64.75±1.29 | 67.27±3.14 | 76.23±3.73 | 75.21 |

Table 15: Full results using classifiers trained by ERM for Table 1 of the manuscript on PACS. We use ResNet-18 as a backbone network.

| Method | A | C | P | S | Avg |
|---|---|---|---|---|---|
| ERM | 77.78±0.81 | 75.09±1.22 | 95.19±0.29 | 69.11±1.22 | 79.29 |
| +Tent | 82.21±1.07 | 81.20±0.51 | 95.32±0.33 | 76.82±1.97 | 83.89 |
| +TentAdapter | 78.89±0.67 | 77.45±0.82 | 95.77±0.40 | 70.89±2.75 | 80.75 |
| +TentClf | 78.16±1.05 | 75.01±1.53 | 95.50±0.35 | 65.60±5.96 | 78.57 |
| +SHOT | 81.09±0.86 | 79.68±0.91 | 96.18±0.27 | 72.48±2.04 | 82.36 |
| +SHOTIM | 81.10±0.90 | 79.66±0.95 | 96.18±0.27 | 72.35±2.03 | 82.33 |
| +PL | 76.42±4.89 | 61.05±5.48 | 95.70±0.56 | 50.75±8.79 | 70.98 |
| +PLClf | 79.09±1.41 | 75.46±2.93 | 95.43±0.32 | 62.48±7.31 | 78.11 |
| +T3A | 78.81±0.97 | 77.14±1.20 | 95.92±0.36 | 71.44±1.63 | 80.83 |
| +TAST (Ours) | 80.56±0.53 | 78.26±0.99 | 96.44±0.20 | 72.52±0.77 | 81.94 |
| +TAST-BN (Ours) | 86.49±0.20 | 83.70±2.57 | 97.23±0.11 | 80.85±1.42 | 87.07 |

Table 16: Full results using classifiers trained by ERM for Table 1 of the manuscript on OfficeHome. We use ResNet-18 as a backbone network.

| Method | A | C | P | R | Avg |
|---|---|---|---|---|---|
| ERM | 55.19±0.49 | 47.76±1.02 | 72.22±0.53 | 73.21±0.89 | 62.10 |
| +Tent | 53.39±0.61 | 48.28±0.88 | 70.50±0.68 | 71.29±0.72 | 60.86 |
| +TentAdapter | 55.53±0.43 | 49.53±0.95 | 72.47±0.27 | 73.01±1.23 | 62.64 |
| +TentClf | 55.17±0.67 | 36.73±1.94 | 72.21±0.52 | 73.22±0.97 | 59.33 |
| +SHOT | 55.14±0.57 | 50.27±1.18 | 71.69±0.45 | 73.21±0.91 | 62.58 |
| +SHOTIM | 55.08±0.56 | 50.29±1.17 | 71.71±0.40 | 73.21±0.90 | 62.57 |
| +PL | 54.49±1.06 | 34.66±13.13 | 71.45±0.37 | 72.20±0.65 | 58.20 |
| +PLClf | 55.14±0.70 | 47.70±1.25 | 72.21±0.54 | 72.62±0.96 | 61.92 |
| +T3A | 55.10±0.74 | 49.56±1.14 | 74.10±0.55 | 74.07±1.18 | 63.21 |
| +TAST (Ours) | 56.15±0.68 | 50.04±1.31 | 74.33±0.28 | 74.28±1.23 | 63.70 |
| +TAST-BN (Ours) | 55.11±0.58 | 51.35±0.85 | 72.58±0.80 | 72.13±0.78 | 62.79 |

Table 17: Full results using classifiers trained by ERM for Table 1 of the manuscript on TerraIncognita. We use ResNet-18 as a backbone network.

| Method | L100 | L38 | L43 | L46 | Avg |
|---|---|---|---|---|---|
| ERM | 37.18±2.46 | 36.12±4.20 | 53.18±1.27 | 36.02±1.37 | 40.62 |
| +Tent | 38.29±0.48 | 25.82±3.91 | 41.53±1.59 | 29.15±1.83 | 33.70 |
| +TentAdapter | 40.55±1.46 | 37.44±2.22 | 46.33±1.32 | 35.30±1.26 | 39.91 |
| +TentClf | 34.44±13.31 | 34.19±5.76 | 52.71±2.03 | 31.86±2.26 | 38.30 |
| +SHOT | 33.87±0.66 | 28.58±2.10 | 40.99±2.07 | 30.83±1.26 | 33.57 |
| +SHOTIM | 33.83±1.29 | 28.13±2.30 | 40.81±2.18 | 30.64±1.46 | 33.35 |
| +PL | 51.92±1.19 | 35.61±20.74 | 39.97±10.98 | 22.26±8.21 | 37.44 |
| +PLClf | 45.22±2.45 | 36.03±5.81 | 52.76±1.54 | 33.10±2.27 | 41.78 |
| +T3A | 36.22±1.89 | 40.08±1.98 | 50.72±1.02 | 33.79±1.25 | 40.20 |
| +TAST (Ours) | 43.67±2.83 | 39.24±3.79 | 52.64±3.02 | 35.01±1.09 | 42.64 |
| +TAST-BN (Ours) | 51.06±7.31 | 32.74±7.54 | 41.70±2.86 | 32.21±3.05 | 39.43 |

Table 18: Full results using classifiers trained by ERM for Table 1 of the manuscript on VLCS. We use ResNet-50 as a backbone network.

| Method | C | L | S | V | Avg |
|---|---|---|---|---|---|
| ERM | 97.66±0.64 | 63.87±1.71 | 71.21±1.52 | 74.09±2.06 | 76.71 |
| +Tent | 92.36±2.44 | 58.46±3.29 | 67.84±2.03 | 73.19±2.68 | 72.96 |
| +TentAdapter | 85.36±3.49 | 58.35±3.46 | 66.47±2.71 | 68.42±2.11 | 69.65 |
| +TentClf | 97.61±0.58 | 63.67±2.10 | 68.77±1.27 | 73.16±1.31 | 75.80 |
| +SHOT | 98.72±1.50 | 46.82±2.57 | 55.70±1.78 | 67.04±2.88 | 67.07 |
| +SHOTIM | 98.65±1.46 | 46.54±2.32 | 55.81±2.32 | 66.73±2.82 | 66.93 |
| +PL | 98.48±0.34 | 53.45±2.82 | 59.45±9.24 | 66.24±8.63 | 69.41 |
| +PLClf | 97.63±0.64 | 63.36±2.10 | 69.74±0.78 | 71.86±4.53 | 75.65 |
| +T3A | 99.17±0.38 | 64.78±1.61 | 73.01±3.24 | 72.20±2.84 | 77.29 |
| +TAST (Ours) | 99.35±0.30 | 65.64±1.78 | 73.63±3.58 | 72.01±2.68 | 77.66 |
| +TAST-BN (Ours) | 96.09±2.40 | 60.22±6.08 | 65.78±6.51 | 71.99±5.90 | 73.52 |

Table 19: Full results using classifiers trained by ERM for Table 1 of the manuscript on PACS. We use ResNet-50 as a backbone network.

| Method | A | C | P | S | Avg |
|---|---|---|---|---|---|
| ERM | 82.92±1.65 | 78.05±3.36 | 96.50±0.32 | 75.38±3.31 | 83.21 |
| +Tent | 82.54±1.32 | 84.90±1.35 | 95.45±0.93 | 77.74±1.36 | 85.16 |
| +TentAdapter | 82.75±2.01 | 79.50±2.26 | 96.78±0.20 | 75.73±3.22 | 83.69 |
| +TentClf | 83.00±1.87 | 77.86±4.20 | 96.55±0.36 | 73.25±6.14 | 82.66 |
| +SHOT | 84.67±1.70 | 80.17±1.39 | 96.58±0.52 | 74.86±2.95 | 84.07 |
| +SHOTIM | 84.62±1.79 | 80.24±1.41 | 96.54±0.46 | 75.16±2.88 | 84.14 |
| +PL | 84.59±5.51 | 76.35±2.57 | 96.41±0.68 | 69.54±11.22 | 81.72 |
| +PLClf | 83.88±2.00 | 78.93±3.68 | 96.53±0.40 | 73.96±6.08 | 83.33 |
| +T3A | 83.56±2.03 | 79.75±3.14 | 96.99±0.24 | 75.36±3.57 | 83.92 |
| +TAST (Ours) | 83.85±2.05 | 79.15±3.03 | 96.93±0.27 | 76.49±3.13 | 84.11 |
| +TAST-BN (Ours) | 87.11±2.04 | 88.50±1.93 | 97.79±0.47 | 83.23±1.42 | 89.16 |

Table 20: Full results using classifiers trained by ERM for Table 1 of the manuscript on OfficeHome. We use ResNet-50 as a backbone network.

| Method | A | C | P | R | Avg |
|---|---|---|---|---|---|
| ERM | 61.32±0.69 | 53.44±1.11 | 75.84±1.10 | 77.90±0.92 | 67.13 |
| +Tent | 60.98±0.67 | 53.94±1.24 | 74.49±0.71 | 75.75±0.53 | 66.29 |
| +TentAdapter | 62.63±0.45 | 54.90±1.17 | 76.20±1.09 | 77.92±1.01 | 67.91 |
| +TentClf | 61.35±0.73 | 52.72±1.40 | 75.23±1.05 | 77.86±1.07 | 66.79 |
| +SHOT | 61.91±0.33 | 55.58±0.91 | 75.49±1.54 | 77.60±0.80 | 67.65 |
| +SHOTIM | 61.84±0.32 | 55.63±0.92 | 75.56±1.60 | 77.57±0.79 | 67.65 |
| +PL | 59.42±1.55 | 42.40±12.31 | 73.80±2.26 | 75.77±1.50 | 62.85 |
| +PLClf | 61.35±0.40 | 52.87±1.96 | 75.86±1.09 | 77.94±1.10 | 67.01 |
| +T3A | 61.91±0.59 | 55.07±1.14 | 77.39±1.38 | 78.67±0.61 | 68.26 |
| +TAST (Ours) | 62.43±0.80 | 55.81±1.26 | 77.46±1.07 | 78.83±0.93 | 68.63 |
| +TAST-BN (Ours) | 63.22±0.85 | 58.20±0.98 | 77.14±1.10 | 76.94±0.39 | 68.88 |

Table 21: Full results using classifiers trained by ERM for Table 1 of the manuscript on TerraIncognita. We use ResNet-50 as a backbone network.

| Method | L100 | L38 | L43 | L46 | Avg |
|---|---|---|---|---|---|
| ERM | 46.84±1.96 | 43.24±2.51 | 53.32±1.92 | 40.30±1.93 | 45.93 |
| +Tent | 41.20±2.71 | 29.72±3.59 | 41.35±2.92 | 36.03±2.85 | 37.08 |
| +TentAdapter | 46.64±1.17 | 41.11±3.16 | 49.31±1.05 | 38.52±2.04 | 43.89 |
| +TentClf | 49.87±3.80 | 43.31±3.19 | 53.01±2.31 | 28.40±6.19 | 43.64 |
| +SHOT | 36.17±2.70 | 29.80±2.92 | 41.00±0.30 | 33.83±1.86 | 35.20 |
| +SHOTIM | 35.56±2.76 | 27.49±4.01 | 40.77±0.45 | 33.67±1.84 | 34.37 |
| +PL | 56.75±5.78 | 46.12±1.03 | 29.44±10.14 | 20.06±4.65 | 38.09 |
| +PLClf | 52.28±3.95 | 43.76±2.96 | 52.78±2.15 | 37.81±2.49 | 46.66 |
| +T3A | 45.13±1.26 | 44.67±2.56 | 52.52±0.78 | 40.13±2.31 | 45.61 |
| +TAST (Ours) | 53.01±3.95 | 43.27±3.21 | 53.79±2.72 | 39.66±3.65 | 47.43 |
| +TAST-BN (Ours) | 55.75±2.37 | 33.92±9.86 | 43.87±4.70 | 32.33±4.40 | 41.47 |

Table 22: Average accuracy(%) using classifiers trained by CORAL on the domain generalization benchmarks for Table 5, namely VLCS, PACS, OfficeHome, and TerraIncognita. We use ResNet-18 and ResNet-50 as backbone networks. **Bold** indicates the best performance for each benchmark. Our proposed method TAST outperforms all the baselines on most of the benchmarks.

| Method | Backbone | VLCS | PACS | OfficeHome | TerraIncognita | Avg |
|---|---|---|---|---|---|---|
| CORAL | | 74.00±1.13 | 81.00±0.79 | 62.78±0.06 | 36.51±2.35 | 63.57 |
| +Tent | | 71.13±1.45 | 84.17±0.61 | 62.37±0.09 | 36.71±0.77 | 63.60 |
| +TentAdapter | | 65.66±1.86 | 82.28±0.36 | 63.37±0.13 | 37.89±1.23 | 62.30 |
| +TentClf | | 72.27±1.29 | 75.71±1.74 | 62.65±0.08 | 30.27±6.34 | 60.23 |
| +SHOT | | 66.01±3.75 | 84.67±0.47 | 63.54±0.23 | 33.20±0.49 | 61.86 |
| +SHOTIM | ResNet-18 | 65.75±3.70 | 84.63±0.49 | 63.53±0.21 | 33.10±0.42 | 61.75 |
| +PL | | 66.58±1.92 | 76.46±3.23 | 61.19±1.52 | 29.32±5.57 | 58.39 |
| +PLClf | | 73.70±0.39 | 76.16±2.44 | 62.68±0.13 | 34.29±3.96 | 61.71 |
| +T3A | | 75.49±1.67 | 82.75±0.51 | 63.72±0.32 | 38.39±1.39 | 65.09 |
| +TAST (Ours) | | 74.82±2.43 | 83.16±0.81 | **64.00±0.25** | **39.21±1.75** | 65.30 |
| +TAST-BN (Ours) | | **77.01±0.36** | **87.21±0.57** | 62.98±0.23 | 37.45±1.11 | **66.16** |
| CORAL | | 76.39±1.01 | 83.52±0.67 | 66.89±0.20 | 42.79±1.27 | 67.40 |
| +Tent | | 74.43±0.98 | 86.50±0.77 | 66.30±0.28 | 42.15±2.81 | 67.35 |
| +TentAdapter | | 68.26±1.39 | 85.05±0.59 | 67.68±0.20 | 41.54±0.93 | 65.63 |
| +TentClf | | 76.45±1.00 | 82.14±1.71 | 64.03±0.56 | 39.74±2.47 | 65.59 |
| +SHOT | | 64.11±0.79 | 85.09±1.03 | 67.73±0.29 | 33.96±0.59 | 62.72 |
| +SHOTIM | ResNet-50 | 63.63±0.60 | 85.06±0.93 | 67.72±0.29 | 34.17±0.90 | 62.65 |
| +PL | | 72.74±1.32 | 75.96±6.46 | 60.74±2.91 | 36.69±3.47 | 61.53 |
| +PLClf | | 75.68±1.23 | 83.56±0.80 | 66.24±0.42 | **44.93±3.76** | 67.60 |
| +T3A | | 77.33±0.97 | 84.54±0.63 | 68.08±0.34 | 43.50±0.19 | 68.36 |
| +TAST (Ours) | | 77.23±1.25 | 85.04±0.49 | 68.39±0.54 | 44.22±1.33 | 68.72 |
| +TAST-BN (Ours) | | **79.13±0.43** | **90.41±0.64** | **69.04±0.36** | 43.46±4.46 | **70.51** |

Table 23: Full results using classifiers trained by CORAL for Table 22 on VLCS. We use ResNet-18 as a backbone network.

| Method | C | L | S | V | Avg |
|---|---|---|---|---|---|
| CORAL | 93.31±3.73 | 61.11±1.66 | 70.62±0.87 | 70.95±0.36 | 74.00 |
| +Tent | 95.78±1.20 | 59.24±0.85 | 63.38±2.06 | 66.13±3.09 | 71.13 |
| +TentAdapter | 79.89±5.76 | 54.29±3.98 | 62.72±1.06 | 65.76±1.79 | 65.66 |
| +TentClf | 94.96±2.77 | 58.42±3.37 | 71.01±1.26 | 64.71±5.42 | 72.27 |
| +SHOT | 88.21±11.66 | 50.29±2.91 | 58.00±1.79 | 67.55±0.51 | 66.01 |
| +SHOTIM | 87.74±11.38 | 49.89±2.74 | 57.73±1.76 | 67.61±0.49 | 65.75 |
| +PL | 95.78±1.34 | 54.09±4.31 | 55.03±2.09 | 61.42±9.04 | 66.58 |
| +PLClf | 95.04±2.59 | 57.67±3.74 | 71.00±1.33 | 71.09±0.48 | 73.70 |
| +T3A | 97.10±3.42 | 63.61±3.43 | 67.90±0.78 | 73.34±1.26 | 75.49 |
| +TAST (Ours) | 95.36±7.76 | 62.95±3.23 | 69.06±1.05 | 71.89±1.95 | 74.82 |
| +TAST-BN (Ours) | 98.90±0.58 | 61.01±2.41 | 69.74±2.57 | 78.40±1.15 | 77.01 |

Table 24: Full results using classifiers trained by CORAL for Table 22 on PACS. We use ResNet-18 as a backbone network.

| Method | A | C | P | S | Avg |
|---|---|---|---|---|---|
| CORAL | 78.74±1.79 | 74.57±1.79 | 92.48±0.90 | 78.21±1.51 | 81.00 |
| +Tent | 82.11±0.95 | 81.22±1.02 | 95.15±0.04 | 78.20±1.33 | 84.17 |
| +TentAdapter | 80.06±0.86 | 77.14±1.35 | 94.05±0.31 | 77.87±0.50 | 82.28 |
| +TentClf | 77.14±0.99 | 63.00±6.22 | 92.93±1.12 | 69.77±2.86 | 75.71 |
| +SHOT | 82.92±1.33 | 81.13±1.18 | 95.28±0.60 | 79.37±1.05 | 84.67 |
| +SHOTIM | 82.92±1.24 | 81.06±1.17 | 95.30±0.57 | 79.23±1.10 | 84.63 |
| +PL | 83.44±1.79 | 67.36±9.25 | 94.20±2.49 | 60.82±12.65 | 76.46 |
| +PLClf | 79.72±1.06 | 62.98±8.19 | 93.12±0.60 | 68.81±1.23 | 76.16 |
| +T3A | 80.68±1.15 | 77.52±0.54 | 93.25±0.66 | 79.53±0.70 | 82.75 |
| +TAST (Ours) | 80.88±1.33 | 77.86±0.92 | 94.28±0.56 | 79.60±1.60 | 83.16 |
| +TAST-BN (Ours) | 86.96±0.66 | 83.58±1.32 | 96.59±0.62 | 81.69±1.16 | 87.21 |

Table 25: Full results using classifiers trained by CORAL for Table 22 on OfficeHome. We use ResNet-18 as a backbone network.

| Method | A | C | P | R | Avg |
|---|---|---|---|---|---|
| CORAL | 55.78±0.29 | 50.09±0.09 | 72.09±0.32 | 73.16±0.39 | 62.78 |
| +Tent | 55.33±0.33 | 50.79±0.31 | 71.08±0.32 | 72.29±0.41 | 62.37 |
| +TentAdapter | 56.59±0.34 | 51.54±0.07 | 72.19±0.19 | 73.17±0.30 | 63.37 |
| +TentClf | 55.56±0.40 | 49.96±0.16 | 72.07±0.44 | 73.00±0.44 | 62.65 |
| +SHOT | 55.77±0.29 | 52.67±0.44 | 72.22±0.64 | 73.50±0.28 | 63.54 |
| +SHOTIM | 55.72±0.28 | 52.64±0.39 | 72.23±0.63 | 73.52±0.29 | 63.53 |
| +PL | 54.17±1.83 | 46.74±4.73 | 71.53±0.40 | 72.34±0.89 | 61.19 |
| +PLClf | 55.77±0.38 | 50.04±0.44 | 71.90±0.37 | 73.03±0.42 | 62.68 |
| +T3A | 55.83±0.36 | 51.68±0.51 | 73.70±0.43 | 73.66±0.55 | 63.72 |
| +TAST (Ours) | 56.22±0.57 | 51.73±0.48 | 74.05±0.73 | 74.00±0.52 | 64.00 |
| +TAST-BN (Ours) | 54.71±0.40 | 52.07±0.61 | 72.80±0.65 | 72.33±0.49 | 62.98 |

Table 26: Full results using classifiers trained by CORAL for Table 22 on TerraIncognita. We use ResNet-18 as a backbone network.

| Method | L100 | L38 | L43 | L46 | Avg |
|---|---|---|---|---|---|
| CORAL | 38.41±2.79 | 25.98±6.65 | 45.59±2.53 | 36.08±1.75 | 36.51 |
| +Tent | 37.31±3.54 | 24.76±1.90 | 45.99±0.73 | 38.79±1.78 | 36.71 |
| +TentAdapter | 41.76±1.23 | 33.9±2.32 | 40.59±3.21 | 35.29±0.98 | 37.89 |
| +TentClf | 29.99±12.66 | 15.84±18.76 | 43.36±4.29 | 31.89±2.89 | 30.27 |
| +SHOT | 35.95±0.88 | 25.85±1.40 | 38.33±0.69 | 32.67±0.83 | 33.20 |
| +SHOTIM | 35.81±0.77 | 25.64±1.15 | 38.16±0.75 | 32.78±0.45 | 33.10 |
| +PL | 37.32±23.49 | 24.27±24.17 | 31.84±7.57 | 23.84±6.83 | 29.32 |
| +PLClf | 45.07±7.63 | 20.52±19.27 | 44.4±1.32 | 27.18±3.00 | 34.29 |
| +T3A | 37.14±2.17 | 34.49±3.47 | 45.00±3.91 | 36.91±1.86 | 38.39 |
| +TAST (Ours) | 46.01±2.18 | 32.11±4.30 | 43.31±3.17 | 35.42±2.53 | 39.21 |
| +TAST-BN (Ours) | 43.04±3.00 | 32.25±5.90 | 42.53±2.54 | 31.99±0.81 | 37.45 |

Table 27: Full results using classifiers trained by CORAL for Table 22 on VLCS. We use ResNet-50 as a backbone network.

| Method | C | L | S | V | Avg |
|---|---|---|---|---|---|
| CORAL | 96.82±1.06 | 62.51±0.81 | 71.46±1.71 | 74.79±3.23 | 76.39 |
| +Tent | 96.53±2.09 | 59.55±0.71 | 67.96±3.68 | 73.69±2.62 | 74.43 |
| +TentAdapter | 84.24±2.39 | 56.07±2.77 | 63.90±1.46 | 68.84±2.75 | 68.26 |
| +TentClf | 96.99±1.16 | 61.48±1.68 | 72.32±1.97 | 75.00±3.07 | 76.45 |
| +SHOT | 85.07±1.63 | 46.27±2.62 | 56.77±1.03 | 68.34±1.01 | 64.11 |
| +SHOTIM | 83.57±0.89 | 45.86±2.93 | 56.55±0.81 | 68.53±0.98 | 63.63 |
| +PL | 98.32±0.64 | 53.64±6.70 | 67.76±1.16 | 71.22±5.85 | 72.74 |
| +PLClf | 96.85±1.30 | 58.71±2.76 | 72.20±1.96 | 74.94±3.89 | 75.68 |
| +T3A | 98.24±0.70 | 64.69±1.64 | 73.06±2.04 | 73.34±3.81 | 77.33 |
| +TAST (Ours) | 99.15±0.28 | 64.17±2.37 | 72.32±1.75 | 73.27±3.27 | 77.23 |
| +TAST-BN (Ours) | 99.14±0.45 | 64.95±2.74 | 73.39±0.94 | 79.04±2.19 | 79.13 |

Table 28: Full results using classifiers trained by CORAL for Table 22 on PACS. We use ResNet-50 as a backbone network.

| Method | A | C | P | S | Avg |
|---|---|---|---|---|---|
| CORAL | 84.40±1.36 | 79.88±2.80 | 95.58±0.69 | 74.24±3.09 | 83.52 |
| +Tent | 86.12±1.37 | 85.15±1.85 | 96.28±0.78 | 78.47±1.52 | 86.50 |
| +TentAdapter | 84.34±0.98 | 81.63±2.47 | 96.38±0.50 | 77.86±1.10 | 85.05 |
| +TentClf | 84.68±1.88 | 80.32±2.89 | 95.98±1.01 | 67.59±8.18 | 82.14 |
| +SHOT | 85.75±1.92 | 82.38±2.52 | 96.88±0.92 | 75.37±2.56 | 85.09 |
| +SHOTIM | 85.72±1.82 | 82.38±2.43 | 96.84±0.94 | 75.29±1.96 | 85.06 |
| +PL | 84.87±3.66 | 77.93±3.78 | 96.29±0.73 | 44.74±20.19 | 75.96 |
| +PLClf | 85.43±1.52 | 79.81±3.15 | 96.21±0.91 | 72.78±5.06 | 83.56 |
| +T3A | 84.71±1.96 | 81.30±2.98 | 96.68±0.53 | 75.47±2.36 | 84.54 |
| +TAST (Ours) | 85.74±1.77 | 81.05±2.79 | 96.88±0.49 | 76.48±2.33 | 85.04 |
| +TAST-BN (Ours) | 90.95±1.20 | 86.78±1.39 | 98.17±0.43 | 85.72±0.63 | 90.41 |

Table 29: Full results using classifiers trained by CORAL for Table 22 on OfficeHome. We use ResNet-50 as a backbone network.

| Method | A | C | P | R | Avg |
|---|---|---|---|---|---|
| CORAL | 60.84±1.18 | 53.82±0.69 | 76.06±0.25 | 76.85±1.15 | 66.89 |
| +Tent | 61.11±1.05 | 54.16±0.56 | 73.61±0.66 | 76.31±1.28 | 66.30 |
| +TentAdapter | 61.91±1.17 | 55.73±0.67 | 76.28±0.47 | 76.82±1.09 | 67.68 |
| +TentClf | 54.31±1.96 | 49.96±1.28 | 75.30±0.08 | 76.56±1.64 | 64.03 |
| +SHOT | 61.79±1.61 | 56.93±0.72 | 75.97±0.79 | 76.23±1.08 | 67.73 |
| +SHOTIM | 61.75±1.52 | 56.93±0.79 | 75.94±0.81 | 76.24±0.95 | 67.72 |
| +PL | 54.78±3.96 | 40.71±10.95 | 74.19±0.49 | 73.28±1.81 | 60.74 |
| +PLClf | 59.80±0.85 | 53.40±1.46 | 75.60±0.17 | 76.15±1.39 | 66.24 |
| +T3A | 61.59±1.51 | 55.57±0.85 | 77.45±0.74 | 77.72±1.36 | 68.08 |
| +TAST (Ours) | 62.02±1.30 | 55.88±0.86 | 78.06±1.01 | 77.60±1.41 | 68.39 |
| +TAST-BN (Ours) | 63.45±1.48 | 59.01±1.14 | 76.68±0.74 | 77.00±1.13 | 69.04 |

Table 30: Full results using classifiers trained by CORAL for Table 22 on TerraIncognita. We use ResNet-50 as a backbone network.

| Method | L100 | L38 | L43 | L46 | Avg |
|---|---|---|---|---|---|
| CORAL | 45.52±0.53 | 39.33±2.81 | 48.98±2.28 | 37.35±2.71 | 42.79 |
| +Tent | 46.72±2.57 | 34.14±3.01 | 48.05±10.46 | 39.70±3.14 | 42.15 |
| +TentAdapter | 48.27±2.51 | 38.07±1.08 | 44.89±2.73 | 34.92±3.07 | 41.54 |
| +TentClf | 42.57±5.73 | 38.35±3.37 | 44.10±2.18 | 33.92±5.58 | 39.74 |
| +SHOT | 38.95±1.35 | 26.99±1.76 | 40.93±2.51 | 28.98±4.77 | 33.96 |
| +SHOTIM | 39.52±2.40 | 26.45±3.94 | 41.07±2.21 | 29.64±4.16 | 34.17 |
| +PL | 51.29±12.49 | 47.88±5.50 | 29.58±10.72 | 17.99±10.59 | 36.69 |
| +PLClf | 53.69±5.34 | 43.78±6.46 | 47.67±6.30 | 34.57±5.18 | 44.93 |
| +T3A | 45.57±1.04 | 40.31±1.50 | 49.81±2.10 | 38.31±1.60 | 43.50 |
| +TAST (Ours) | 53.81±0.58 | 39.99±2.38 | 48.07±3.86 | 35.02±3.12 | 44.22 |
| +TAST-BN (Ours) | 54.80±3.22 | 41.56±6.79 | 47.09±5.58 | 30.38±4.60 | 43.46 |

Table 31: Average accuracy(%) using classifiers trained by MMD for Table 5 on the domain generalization benchmarks, namely VLCS, PACS, OfficeHome, and TerraIncognita. We use ResNet-18 as a backbone network.

| Method | VLCS | PACS | OfficeHome | TerraIncognita | Avg |
|---|---|---|---|---|---|
| MMD | 74.90±0.50 | 81.06±0.92 | 62.20±0.48 | 35.73±2.70 | 63.47 |
| +Tent | 74.59±0.66 | 84.47±0.19 | 62.15±0.33 | 36.11±0.72 | 64.33 |
| +TentAdapter | 65.94±1.91 | 82.44±0.26 | 62.80±0.46 | 37.66±0.34 | 62.21 |
| +TentClf | 74.91±0.60 | 57.76±5.32 | 62.10±0.50 | 28.94±6.56 | 55.93 |
| +SHOT | 65.41±2.03 | 85.57±0.28 | 63.25±0.53 | 32.34±0.68 | 61.64 |
| +SHOTIM | 65.00±2.20 | 85.53±0.29 | 63.20±0.56 | 32.25±0.55 | 61.50 |
| +PL | 66.17±1.73 | 74.42±4.09 | 60.49±1.03 | 22.30±8.42 | 55.85 |
| +PLClf | 74.84±0.50 | 69.95±0.95 | 62.16±0.45 | 31.94±5.67 | 59.72 |
| +T3A | 77.28±0.45 | 82.52±0.53 | 63.34±0.55 | 37.40±1.86 | 65.14 |
| +TAST (Ours) | 76.21±0.79 | 83.29±0.26 | 63.49±0.49 | 38.12±2.47 | 65.28 |
| +TAST-BN (Ours) | 76.06±0.89 | 86.35±0.76 | 63.22±0.26 | 39.46±1.63 | 66.27 |

Table 32: Full results using classifiers trained by MMD for Table 31 on VLCS. We use ResNet-18 as a backbone network.

| Method | C | L | S | V | Avg |
|---|---|---|---|---|---|
| MMD | 95.51±2.59 | 62.32±0.68 | 70.09±0.80 | 71.68±0.83 | 74.90 |
| +Tent | 97.20±0.55 | 61.53±0.60 | 69.32±2.58 | 70.33±0.92 | 74.59 |
| +TentAdapter | 80.71±5.21 | 54.94±2.31 | 61.88±0.30 | 66.25±1.58 | 65.94 |
| +TentClf | 95.71±2.16 | 61.87±1.02 | 70.15±0.91 | 71.90±0.82 | 74.91 |
| +SHOT | 88.10±8.59 | 48.99±2.74 | 57.14±1.13 | 67.43±0.47 | 65.41 |
| +SHOTIM | 86.93±8.90 | 48.71±2.70 | 56.98±1.17 | 67.37±0.48 | 65.00 |
| +PL | 95.63±1.07 | 51.75±2.10 | 59.89±11.04 | 57.40±6.85 | 66.17 |
| +PLClf | 95.69±2.19 | 61.65±0.85 | 70.14±0.86 | 71.87±0.79 | 74.84 |
| +T3A | 99.15±0.52 | 64.19±0.76 | 71.16±0.74 | 74.62±1.63 | 77.28 |
| +TAST (Ours) | 99.29±0.46 | 63.14±0.66 | 69.62±1.23 | 72.81±1.30 | 76.21 |
| +TAST-BN (Ours) | 99.14±0.48 | 61.54±2.74 | 65.03±1.95 | 78.53±0.35 | 76.06 |

Table 33: Full results using classifiers trained by MMD for Table 31 on PACS. We use ResNet-18 as a backbone network.

| Method | A | C | P | S | Avg |
|---|---|---|---|---|---|
| MMD | 79.52±0.50 | 74.66±1.28 | 93.46±0.95 | 76.60±2.62 | 81.06 |
| +Tent | 82.84±0.97 | 81.22±0.85 | 95.49±0.40 | 78.32±0.48 | 84.47 |
| +TentAdapter | 80.51±0.84 | 77.09±0.44 | 94.55±0.33 | 77.59±0.64 | 82.44 |
| +TentClf | 56.53±16.38 | 47.89±6.86 | 93.37±0.67 | 33.26±8.17 | 57.76 |
| +SHOT | 84.50±0.13 | 81.98±0.98 | 95.58±0.22 | 80.19±0.95 | 85.57 |
| +SHOTIM | 84.49±0.20 | 81.94±0.93 | 95.58±0.22 | 80.09±0.89 | 85.53 |
| +PL | 84.59±0.48 | 53.96±10.45 | 95.10±0.18 | 64.04±11.28 | 74.42 |
| +PLClf | 80.04±0.60 | 63.87±8.49 | 93.60±0.57 | 42.28±7.47 | 69.95 |
| +T3A | 81.92±0.41 | 75.35±1.11 | 94.99±0.36 | 77.81±1.62 | 82.52 |
| +TAST (Ours) | 81.76±0.31 | 77.13±0.51 | 95.37±0.70 | 78.91±1.12 | 83.29 |
| +TAST-BN (Ours) | 86.23±1.18 | 82.16±1.93 | 97.46±0.14 | 79.55±1.08 | 86.35 |

Table 34: Full results using classifiers trained by MMD for Table 31 on OfficeHome. We use ResNet-18 as a backbone network.

| Method | A | C | P | R | Avg |
|---|---|---|---|---|---|
| MMD | 54.39±0.86 | 49.71±1.02 | 71.95±0.59 | 72.74±0.41 | 62.20 |
| +Tent | 55.73±0.47 | 49.71±0.20 | 70.75±0.74 | 72.41±0.45 | 62.15 |
| +TentAdapter | 55.34±0.56 | 51.09±1.09 | 72.11±0.46 | 72.68±0.27 | 62.80 |
| +TentClf | 54.39±0.72 | 49.28±1.18 | 71.95±0.66 | 72.77±0.27 | 62.10 |
| +SHOT | 54.83±0.77 | 52.38±0.95 | 72.59±0.71 | 73.18±0.15 | 63.25 |
| +SHOTIM | 54.85±0.85 | 52.27±0.97 | 72.54±0.71 | 73.15±0.17 | 63.20 |
| +PL | 53.26±1.14 | 44.63±3.44 | 71.76±0.68 | 72.33±0.08 | 60.49 |
| +PLClf | 54.44±0.67 | 49.67±0.97 | 71.97±0.65 | 72.57±0.29 | 62.16 |
| +T3A | 54.43±1.26 | 51.25±0.83 | 73.86±0.57 | 73.84±0.47 | 63.34 |
| +TAST (Ours) | 55.17±1.04 | 50.80±1.14 | 74.20±0.57 | 73.78±0.51 | 63.49 |
| +TAST-BN (Ours) | 54.40±0.60 | 52.21±0.38 | 73.30±0.35 | 72.96±0.42 | 63.22 |

Table 35: Full results using classifiers trained by MMD for Table 31 on TerraIncognita. We use ResNet-18 as a backbone network.

| Method | L100 | L38 | L43 | L46 | Avg |
|---|---|---|---|---|---|
| MMD | 34.57±3.77 | 26.57±6.18 | 46.08±2.85 | 35.72±1.17 | 35.73 |
| +Tent | 37.04±3.72 | 25.40±1.81 | 44.82±0.72 | 37.19±1.64 | 36.11 |
| +TentAdapter | 40.04±2.45 | 33.38±2.98 | 42.38±2.80 | 34.84±1.06 | 37.66 |
| +TentClf | 22.92±9.25 | 15.19±15.33 | 45.24±4.15 | 32.39±1.27 | 28.94 |
| +SHOT | 33.79±1.34 | 25.68±2.73 | 37.81±0.67 | 32.07±2.71 | 32.34 |
| +SHOTIM | 33.48±1.14 | 25.77±1.95 | 37.87±0.61 | 31.87±2.44 | 32.25 |
| +PL | 29.45±26.19 | 13.78±20.60 | 27.69±10.39 | 18.27±5.75 | 22.30 |
| +PLClf | 36.96±8.00 | 17.63±18.15 | 46.28±4.42 | 26.87±2.41 | 31.94 |
| +T3A | 35.16±2.74 | 34.41±2.73 | 43.83±4.22 | 36.19±1.79 | 37.40 |
| +TAST (Ours) | 44.70±1.82 | 31.97±4.96 | 42.30±4.83 | 33.50±4.40 | 38.12 |
| +TAST-BN (Ours) | 43.81±5.05 | 37.35±2.57 | 44.41±1.55 | 32.28±1.73 | 39.46 |

Table 36: Average accuracy(%) using classifiers trained by Mixup for Table 5 on the domain generalization benchmarks, namely VLCS, PACS, OfficeHome, and TerraIncognita. We use ResNet-18 as a backbone network.

| Method | VLCS | PACS | OfficeHome | TerraIncognita | Avg |
|---|---|---|---|---|---|
| Mixup | 74.97±0.86 | 78.29±0.88 | 61.83±0.88 | 41.04±1.01 | 64.03 |
| +Tent | 72.73±0.41 | 83.88±0.51 | 61.82±0.45 | 39.52±0.36 | 64.49 |
| +TentAdapter | 62.83±0.83 | 81.44±0.27 | 62.82±0.64 | 40.72±1.81 | 61.95 |
| +TentClf | 74.33±0.92 | 68.95±2.86 | 61.45±0.88 | 37.21±4.79 | 60.49 |
| +SHOT | 68.69±0.91 | 84.43±0.39 | 62.81±0.42 | 36.32±0.50 | 63.06 |
| +SHOTIM | 68.31±0.98 | 84.52±0.36 | 62.80±0.43 | 36.03±0.61 | 62.92 |
| +PL | 59.90±2.19 | 68.02±2.43 | 60.66±0.63 | 32.30±6.83 | 55.22 |
| +PLClf | 74.19±0.78 | 70.94±3.00 | 61.67±0.86 | 40.63±4.88 | 61.86 |
| +T3A | 78.43±0.76 | 81.91±0.54 | 63.49±0.86 | 39.89±0.90 | 65.93 |
| +TAST (Ours) | 77.19±0.80 | 82.85±0.36 | 63.83±0.74 | 41.44±1.67 | 66.33 |
| +TAST-BN (Ours) | 76.89±0.86 | 87.14±0.56 | 62.09±0.86 | 42.70±1.90 | 67.21 |

Table 37: Full results using classifiers trained by Mixup for Table 36 on VLCS. We use ResNet-18 as a backbone network.

| Method | C | L | S | V | Avg |
|---|---|---|---|---|---|
| Mixup | 94.73±1.35 | 62.52±0.79 | 69.70±0.89 | 72.93±1.27 | 74.97 |
| +Tent | 95.27±0.47 | 59.84±1.01 | 68.61±1.04 | 67.19±1.62 | 72.73 |
| +TentAdapter | 76.52±1.36 | 52.40±1.05 | 60.00±1.63 | 62.41±2.25 | 62.83 |
| +TentClf | 94.87±1.31 | 61.72±0.73 | 68.44±1.04 | 72.29±2.22 | 74.33 |
| +SHOT | 96.47±3.75 | 51.01±1.22 | 58.83±1.55 | 68.47±0.27 | 68.69 |
| +SHOTIM | 95.63±4.31 | 50.49±1.13 | 58.71±1.71 | 68.41±0.35 | 68.31 |
| +PL | 94.94±1.82 | 49.88±0.35 | 45.87±7.30 | 48.90±4.44 | 59.90 |
| +PLClf | 94.82±1.29 | 59.16±0.65 | 69.85±0.94 | 72.92±1.54 | 74.19 |
| +T3A | 99.11±0.76 | 64.75±1.02 | 72.69±2.16 | 77.16±0.79 | 78.43 |
| +TAST (Ours) | 98.97±0.87 | 63.19±0.77 | 71.04±1.80 | 75.56±1.03 | 77.19 |
| +TAST-BN (Ours) | 99.27±0.32 | 60.76±4.75 | 69.63±2.58 | 77.91±1.76 | 76.89 |

Table 38: Full results using classifiers trained by Mixup for Table 36 on PACS. We use ResNet-18 as a backbone network.

| Method | A | C | P | S | Avg |
|---|---|---|---|---|---|
| Mixup | 80.28±2.31 | 70.69±1.19 | 94.26±0.85 | 67.94±0.73 | 78.29 |
| +Tent | 81.51±0.82 | 79.49±1.01 | 95.58±0.18 | 78.94±1.04 | 83.88 |
| +TentAdapter | 82.40±1.00 | 75.98±0.79 | 94.63±0.45 | 72.75±1.57 | 81.44 |
| +TentClf | 79.66±3.73 | 64.24±4.07 | 94.17±1.01 | 37.75±13.33 | 68.95 |
| +SHOT | 85.20±0.56 | 80.13±0.88 | 96.20±0.75 | 76.18±0.45 | 84.43 |
| +SHOTIM | 85.17±0.56 | 80.64±0.92 | 96.22±0.77 | 76.05±0.50 | 84.52 |
| +PL | 82.82±2.54 | 66.48±8.08 | 95.40±0.64 | 27.37±5.71 | 68.02 |
| +PLClf | 79.32±2.88 | 70.61±1.26 | 94.26±0.89 | 39.58±15.47 | 70.94 |
| +T3A | 83.06±1.31 | 75.92±0.58 | 95.87±0.66 | 72.80±2.09 | 81.91 |
| +TAST (Ours) | 83.93±0.99 | 76.75±1.32 | 96.34±0.63 | 74.39±1.69 | 82.85 |
| +TAST-BN (Ours) | 86.18±0.49 | 82.69±1.44 | 97.27±0.48 | 82.43±1.80 | 87.14 |

Table 39: Full results using classifiers trained by Mixup for Table 36 on OfficeHome. We use ResNet-18 as a backbone network.

| Method | A | C | P | R | Avg |
|---|---|---|---|---|---|
| Mixup | 53.92±1.21 | 49.17±1.49 | 71.66±0.52 | 72.56±0.82 | 61.83 |
| +Tent | 53.22±0.92 | 50.75±0.79 | 71.22±0.74 | 72.10±0.62 | 61.82 |
| +TentAdapter | 54.78±1.29 | 51.47±1.09 | 72.21±0.25 | 72.82±0.74 | 62.82 |
| +TentClf | 53.87±1.20 | 48.22±1.48 | 71.65±0.42 | 72.06±0.84 | 61.45 |
| +SHOT | 53.95±1.17 | 52.20±1.27 | 72.29±0.36 | 72.79±0.43 | 62.81 |
| +SHOTIM | 53.94±1.22 | 52.15±1.30 | 72.28±0.42 | 72.83±0.41 | 62.80 |
| +PL | 52.82±0.96 | 48.17±1.38 | 70.97±0.82 | 70.70±1.00 | 60.66 |
| +PLClf | 53.85±1.10 | 49.12±1.43 | 71.64±0.47 | 72.06±1.04 | 61.67 |
| +T3A | 54.83±1.49 | 50.97±1.51 | 74.14±0.63 | 74.00±0.31 | 63.49 |
| +TAST (Ours) | 54.97±0.96 | 51.31±1.84 | 74.88±0.66 | 74.16±0.22 | 63.83 |
| +TAST-BN (Ours) | 53.09±1.44 | 51.04±1.81 | 72.44±0.64 | 71.80±0.64 | 62.09 |

Table 40: Full results using classifiers trained by Mixup for Table 36 on TerraIncognita. We use ResNet-18 as a backbone network.

| Method | L100 | L38 | L43 | L46 | Avg |
|---|---|---|---|---|---|
| Mixup | 52.26±1.44 | 35.90±4.46 | 41.23±2.04 | 34.77±2.14 | 41.04 |
| +Tent | 41.35±2.74 | 33.05±2.75 | 44.17±1.50 | 39.51±1.83 | 39.52 |
| +TentAdapter | 45.35±2.22 | 44.17±2.15 | 42.25±1.98 | 31.11±2.43 | 40.72 |
| +TentClf | 49.62±7.17 | 37.29±12.05 | 37.45±2.99 | 24.48±0.73 | 37.21 |
| +SHOT | 42.78±3.48 | 31.04±2.88 | 39.58±1.29 | 31.86±3.47 | 36.32 |
| +SHOTIM | 42.14±3.38 | 30.71±3.18 | 39.37±1.34 | 31.89±3.51 | 36.03 |
| +PL | 52.56±0.16 | 34.80±21.65 | 23.33±4.85 | 18.51±6.10 | 32.30 |
| +PLClf | 53.30±1.59 | 35.52±18.76 | 40.68±1.15 | 33.04±2.57 | 40.63 |
| +T3A | 43.05±2.28 | 38.53±2.37 | 43.32±3.33 | 34.65±1.22 | 39.89 |
| +TAST (Ours) | 57.57±5.18 | 36.43±3.72 | 38.34±2.38 | 33.40±2.08 | 41.44 |
| +TAST-BN (Ours) | 56.81±4.34 | 42.44±2.03 | 41.01±2.50 | 30.54±1.52 | 42.70 |

Table 41: Full results using classifiers trained by ERM for Table 2 of manuscript on VLCS. We use ResNet-18 as a backbone network.

| Method | $N_e$ | C | L | S | V | Avg |
|---|---|---|---|---|---|---|
| ERM | - | 94.70±1.33 | 63.79±1.30 | 67.90±1.97 | 73.15±1.37 | 74.88 |
| +T3A | - | 97.52±1.99 | 65.32±2.24 | 70.70±3.48 | 75.51±1.75 | 77.26 |
| +TAST-N (Ours) | - | 95.31±4.33 | 65.62±1.79 | 68.9±3.22 | 74.96±1.66 | 76.20 |
| +TAST (Ours) | 1 | 98.62±1.06 | 62.61±1.90 | 66.84±3.01 | 72.73±1.17 | 75.20 |
| +TAST (Ours) | 5 | 99.15±0.68 | 65.58±3.08 | 67.53±1.49 | 74.46±1.87 | 76.68 |
| +TAST (Ours) | 10 | 99.22±0.45 | 66.21±1.36 | 68.62±2.39 | 75.66±2.03 | 77.43 |
| +TAST (Ours) | 20 | 99.17±0.60 | 65.87±1.90 | 68.13±1.76 | 75.92±1.75 | 77.27 |

Table 42: Full results using classifiers trained by ERM for Table 2 of manuscript on PACS. We use ResNet-18 as a backbone network.

| Method | $N_e$ | A | C | P | S | Avg |
|---|---|---|---|---|---|---|
| ERM | - | 77.78±0.81 | 75.09±1.22 | 95.19±0.29 | 69.11±1.22 | 79.29 |
| +T3A | - | 78.81±0.97 | 77.14±1.20 | 95.92±0.36 | 71.44±1.63 | 80.83 |
| +TAST-N (Ours) | - | 80.18±0.88 | 77.34±1.38 | 96.57±0.27 | 72.38±0.77 | 81.62 |
| +TAST (Ours) | 1 | 80.21±0.80 | 77.06±1.44 | 96.13±0.53 | 71.51±1.22 | 81.23 |
| +TAST (Ours) | 5 | 80.85±0.86 | 77.91±0.71 | 96.49±0.37 | 72.00±1.16 | 81.81 |
| +TAST (Ours) | 10 | 79.95±1.42 | 78.12±0.98 | 96.51±0.25 | 71.65±2.32 | 81.56 |
| +TAST (Ours) | 20 | 80.56±0.53 | 78.26±0.99 | 96.44±0.20 | 72.52±0.77 | 81.94 |

Table 43: Full results using classifiers trained by ERM for Table 2 of manuscript on OfficeHome. We use ResNet-18 as a backbone network.

| Method | $N_e$ | A | C | P | R | Avg |
|---|---|---|---|---|---|---|
| ERM | - | 55.19±0.49 | 47.76±1.02 | 72.22±0.53 | 73.21±0.89 | 62.10 |
| +T3A | - | 55.10±0.74 | 49.56±1.14 | 74.10±1.18 | 74.07±1.18 | 63.21 |
| +TAST-N (Ours) | - | 55.25±0.97 | 50.45±1.03 | 74.24±0.55 | 74.23±1.37 | 63.54 |
| +TAST (Ours) | 1 | 53.53±0.90 | 49.46±1.39 | 72.84±0.69 | 72.53±1.30 | 62.09 |
| +TAST (Ours) | 5 | 55.34±1.04 | 50.51±1.03 | 74.23±0.43 | 73.97±0.95 | 63.51 |
| +TAST (Ours) | 10 | 55.76±0.68 | 49.52±1.38 | 74.17±0.49 | 74.11±1.00 | 63.39 |
| +TAST (Ours) | 20 | 56.15±0.68 | 50.04±1.31 | 74.33±0.28 | 74.28±1.23 | 63.70 |

Table 44: Full results using classifiers trained by ERM for Table 2 of manuscript on TerraIncognita. We use ResNet-18 as a backbone network.

| Method | $N_e$ | L100 | L38 | L43 | L46 | Avg |
|---|---|---|---|---|---|---|
| ERM | - | 37.18±2.46 | 36.12±4.20 | 53.18±1.27 | 36.02±1.37 | 40.62 |
| +T3A | - | 36.22±1.89 | 40.08±1.98 | 50.72±1.02 | 33.79±1.25 | 40.20 |
| +TAST-N (Ours) | - | 39.75±1.76 | 39.20±2.65 | 52.33±2.63 | 36.24±1.28 | 41.88 |
| +TAST (Ours) | 1 | 43.23±0.87 | 42.49±2.08 | 51.22±3.69 | 33.41±1.05 | 42.59 |
| +TAST (Ours) | 5 | 43.95±2.33 | 38.89±2.42 | 52.44±3.04 | 35.42±1.27 | 42.68 |
| +TAST (Ours) | 10 | 43.96±2.92 | 38.48±3.56 | 53.27±2.73 | 34.67±1.24 | 42.60 |
| +TAST (Ours) | 20 | 43.67±2.83 | 39.24±3.79 | 52.64±3.02 | 35.01±1.09 | 42.64 |

Table 45: Average error rate (%) on CIFAR-10C for the highest severe corruptions. **Bold** indicates the best performance for each image corruption.

| Method | gauss | brit | contr | defoc | elast | fog | frost | glass | impul | jpeg | motn | pixel | shot | snow | zoom |
|---|---|---|---|---|---|---|---|---|---|---|---|---|---|---|---|
| No adaptation | 48.73 | **7.01** | 13.27 | 11.84 | 23.38 | 29.41 | 28.24 | 50.78 | 57.00 | 19.46 | 23.38 | 47.88 | 44.00 | 21.93 | 10.84 |
| +SHOT | 17.09 | 8.64 | 8.57 | 9.83 | 19.53 | 19.72 | 13.93 | 25.60 | 27.15 | 13.98 | 14.01 | 11.68 | 16.02 | 15.89 | 8.22 |
| +Tent | 15.91 | 7.91 | 7.85 | 9.27 | 18.13 | 16.45 | 12.62 | 23.48 | 24.52 | 13.19 | 12.70 | 10.93 | 14.59 | 14.06 | 7.68 |
| +PL | 33.56 | 7.54 | 11.53 | 10.60 | 20.21 | 23.86 | 21.78 | 38.36 | 43.64 | 16.88 | 18.72 | 29.83 | 30.43 | 18.75 | 9.43 |
| +T3A | 41.87 | 7.30 | 13.61 | 11.99 | 22.06 | 28.52 | 27.13 | 44.10 | 54.26 | 18.71 | 22.54 | 37.53 | 37.84 | 21.97 | 10.72 |
| +TAST (Ours) | 42.02 | 7.34 | 13.55 | 11.86 | 21.38 | 28.58 | 26.51 | 44.99 | 54.19 | 18.96 | 22.55 | 37.08 | 37.62 | 21.84 | 10.64 |
| +TAST-BN (Ours) | **14.91** | 7.68 | 7.81 | **8.62** | **16.81** | **15.10** | 12.25 | **21.82** | 22.54 | 12.38 | **11.67** | 10.34 | **13.77** | **12.99** | 7.57 |
| +TTT++ | 16.25 | 7.27 | **7.46** | 9.12 | 18.17 | 17.72 | 12.36 | 25.74 | 26.43 | 13.13 | 12.85 | 11.38 | 15.02 | 14.40 | 7.59 |

Table 46: Average error rate (%) on CIFAR-100C for the highest severe corruptions. **Bold** indicates the best performance for each image corruption.

| Method | gauss | brit | contr | defoc | elast | fog | frost | glass | impul | jpeg | motn | pixel | shot | snow | zoom |
|---|---|---|---|---|---|---|---|---|---|---|---|---|---|---|---|
| No adaptation | 80.77 | **28.86** | 50.93 | 39.62 | 59.54 | 68.11 | 60.19 | 54.79 | 82.26 | 87.75 | 49.96 | 54.22 | 72.27 | 77.84 | 54.58 |
| +SHOT | 45.95 | 30.14 | 31.93 | 32.81 | 46.19 | 49.49 | 40.65 | 54.79 | 57.02 | 37.99 | 39.22 | 37.57 | 44.33 | 44.08 | 30.97 |
| +Tent | 43.02 | 29.65 | 30.52 | 31.48 | 43.88 | 44.03 | 39.21 | 50.91 | 53.10 | 36.22 | 36.31 | 34.10 | 41.58 | 41.85 | 29.73 |
| +PL | 43.94 | 30.14 | 31.20 | 32.11 | 45.07 | 46.57 | 40.11 | 52.66 | 54.48 | 37.48 | 36.92 | 34.59 | 42.68 | 42.77 | 30.19 |
| +T3A | 76.95 | 29.54 | 48.02 | 39.64 | 55.68 | 65.90 | 58.45 | 78.23 | 86.39 | 48.82 | 53.46 | 66.31 | 74.14 | 55.01 | 37.68 |
| +TAST (Ours) | 80.13 | 29.40 | 50.86 | 40.43 | 58.13 | 69.24 | 60.89 | 81.94 | 88.94 | 50.44 | 57.26 | 70.58 | 77.47 | 56.98 | 38.46 |
| +TAST-BN (Ours) | **42.01** | 29.00 | **30.20** | **30.74** | **42.97** | **41.02** | **38.19** | **48.95** | **51.20** | **35.70** | **35.03** | **33.38** | **40.01** | **39.88** | **29.07** |
| +TTT++ | 47.10 | 29.99 | 31.10 | 32.61 | 47.73 | 51.74 | 41.37 | 57.36 | 60.40 | 38.93 | 39.01 | 37.17 | 45.34 | 44.53 | 31.31 |

