# OpenReview forum: "Test-Time Adaptation via Self-Training with Nearest Neighbor Information"
_ICLR.cc/2023/Conference — ICLR 2023 poster_

### Official Review · Reviewer_tB9n · 2022-10-21

**Confidence:** 4
**Correctness:** 3
**Technical Novelty And Significance:** 3
**Empirical Novelty And Significance:** 3
**Recommendation:** 6

**Clarity, Quality, Novelty And Reproducibility:**

The proposed approach builds upon T3A but has sufficient novelty. Quality of the empirical evaluation is good, albeit I would ask the authors to also conduct an evaluation on ImageNet-C to demonstrate that TAST can also deal with many classes (1000). Reproducibility of the approach is an open question to me since no code is provided and clarity of presentation is missing. I list several issues in terms of clarity below:
 - In Equation 2, the same symbol $\mathcal{S}^k_t$ refers to two different sets on the left-hand and right-hand side of equation (after and before filtering high entropy samples)
 - Which distance function $d$ is used in Equation 4?
 - The set $\mathcal{S}$ is a set of class-specific sets $\mathcal{S}^K$, but the authors use in Eq. 3 and later the notation $z \in \mathcal{S}$ as if it would be a flat set of feature vectors. This is confusing as it is not clear how this flattened set is obtained (just as the union of al sets $\mathcal{S}^K$?)
 - The adaptation modules $h_{\phi_i}$ are not defined properly on the bottom of page 4, e.g. what is their output dimensionality?
 - For the cross-entropy based loss $\mathcal{L}^{TAST}$ it is not clear if the gradient wrt. $\phi$ is propagated into both arguments of the cross-entropy or only one of them? In any case, it would be worth investigating which of the two options is preferable.


Minor:
 - T3A should be cited again on the page 3
 - The statement "Although it achieves effective test-time adaptation, it has a limitation that it can be utilized only if there are BN layers in the trained classifier." is not quite true, because one can always add affine layer  anywhere in a pretrained classifier, initialize them as identity and then adapt during TTA.
 - I would recommend to move the related works section before Section 2, as it will introduce several baseline methods used in Section 4.

**Strength And Weaknesses:**

Strength:
 * The paper is overall well written and presented
 * The proposed TAST approach is a novel contribution (extending T3A)
 * Figure 1 provides a useful illustration of the proposed procedure
 * The empirical evaluation on domain generalization benchmarks is extensive and strong performance of the proposed TAST is demonstrated.
 * Reasonable ablation studies on TAST are conducted.

Weaknesses:
 * Clarity is insufficient, mostly because there are several inconsistencies in notation and undefined symbols (see below)
 * The ensembling of adaptation modules is not well motivated and also the trade-off such an ensemble poses in terms of increasing inference time is not discussed or quantified. Also, if one employs an ensemble, wouldn't it be natural/preferable to also employ different support sets in conjunction with different adaptation models in order to increase diversity within the ensemble?
 * The method TAST has several additional hyperparameters (N_s, T, M, N_e) and it is not entirely clear how these have been chosen in certain experiments. In particular, they should not be chosen in any way based on the target domains. In general, TTA methods should have few if any free hyperparameters, so it would be preferable to clearly state default values for hyperparameters and use these everywhere except for ablation studies.
 * Experiments on TTA on common corruptions are only conducted on CIFAR10/100-C and not ImageNet-C. Is this because 1000 classes are a challenging for TAST?
 * Runtime of different TTA methods is not reported but a relevant quantity when considering TTA in practical applications. Specifically, would TAST-BN, T3a be slower or faster than TAST?
 * The weak performance of TAST on CIFAR10/100-C is not well explained. A summary sentence like "T3A and TAST show slight performance changes compared to existing TTA methods that fine-tune the feature extractor." does not reflect well that TAST/T3A are more than 20 percent points worse than TENT on CIFAR100-C. I think it is fair to say that CIFAR10/100-C require adapting the entire feature extractor and thus, only TAST-BN is a competitive method here, while TAST is not performing systematically better than "No Adaptation".


**Summary Of The Paper:**

The paper proposes a new approach for fully test-time adaptation. The approach aims to prevent model collapse due to confirmation bias by using nearest neighbour information for pseudo-labelling. It then adapts an ensemble of adaptation modules that are used instead of the original linear classifier layer, on top of the feature representation. The approach is evaluated on four domain generalization benchmarks and on the common corruptions benchmark on CIFAR-10/100.

**Summary Of The Review:**

Overall, in the current state I find clarity of the exposition lacking and would tend towards rejection. However, I would be willing to reconsider my evaluation if the issues mentioned above would be addressed.

### Update after discussion period ###
The authors have devoted considerable work in clarifying open questions and revising the manuscript. Overall, I think the proposed TAST method can be useful in certain settings (image classification problems with few classes, high level semantic shifts) but in others, like (a) image corruptions (where only TAST-BN performs competitive) or (b) with many classes, it remains preferable to adapt the feature extractor/backbone itself. I attribute (a) to lower-level perceptual shifts being handled better by earlier features and (b) to the property that features unlike prototypes are shared over classes and thus adapting them is more data-efficient. I think the remaining "niche" for TAST is still useful and relevant but I would like the authors to be more transparent about the limitations and avoid broad statements like "TAST outperforms the state-of-the-art TTA methods" in the abstract. Assuming the discussion of strengths and weaknesses becomes more balanced in the final version, I would lean towards acceptance and I increase my score accordingly.

---

> ### Author Response · Authors · 2022-11-16
> **Response to Reviewer tB9n (3/3)**
>
> > W2/W5. the trade-off such an ensemble poses in terms of increasing inference time is not discussed or quantified/ Runtime of different TTA methods is not reported but a relevant quantity when considering TTA in practical applications. Specifically, would TAST-BN, T3a be slower or faster than TAST?
>
> We reported an average runtime spent to adapt classifiers during test time in Table 4 of Appendix A. We additionally report (1) the number of learnable parameters during adaptation and (2) an average runtime to adapt trained classifiers with a single hyperparameter combination for all test-time adaptation algorithms in Table R1-2 of “Common Response to Reviewers”.
>
> > W6. The weak performance of TAST on CIFAR10/100-C is not well explained. A summary sentence like "T3A and TAST show slight performance changes compared to existing TTA methods that fine-tune the feature extractor." does not reflect well that TAST/T3A are more than 20 percent points worse than TENT on CIFAR100-C. I think it is fair to say that CIFAR10/100-C require adapting the entire feature extractor and thus, only TAST-BN is a competitive method here, while TAST is not performing systematically better than "No Adaptation".
>
> We agree with the reviewer that an adapted feature extractor is required in image corruption benchmarks and thus only TAST-BN is a competitive method compared to the existing test-time adaptation methods. We will modify the comments in the manuscript.
>
> > Which distance function $d$ is used in Equation 4?
>
> We use the cosine similarity as a distance metric $d$ as commented in the footnote of page 3.
>
> > The set $\mathcal{S}$ is a set of class-specific sets $\mathcal{S}_K$, but the authors use in Eq. 3 and later the notation $z \in \mathcal{S}$ as if it would be a flat set of feature vectors. This is confusing as it is not clear how this flattened set is obtained (just as the union of al sets $\mathcal{S}_K$?)
>
> Yes. $\mathcal{S}$ is the union of the sets $\mathcal{S}_K$ as described in the middle of page 3.
>
> > The adaptation modules $h_\phi$ are not defined properly on the bottom of page 4, e.g. what is their output dimensionality?
>
> As described in "Training setup" paragraph on page 7, we use BatchEnsemble for the adaptation modules for TAST and the output dimension of each adaptation module is set to 1/4 of the output dimension of the feature extractor. Moreover, we showed the performance of TAST is robust to changes in the output dimension in Appendix C.  We modify the manuscript to indicate that the detailed explanation of the adaptation modules is described in Section 4.1.1. and Appendix A.
>
> > For the cross-entropy based loss $L_{\text{TAST}}$, it is not clear if the gradient wrt. $\phi$ is propagated into both arguments of the cross-entropy or only one of them? In any case, it would be worth investigating which of the two options is preferable.
>
> Our method does not propagate gradients into the pseudo labels as is standard in self-training methods [C,D,E,F,G]. To avoid the confusion, we clarify it in Section 3.
>
> [C] Lee et al.,  Pseudo-label : The simple and efficient semi-supervised learning method for deep neural networks. ICML 2013 Workshop : Challenges in Representation Learning.
>
> [D] Sohn et al., Fixmatch: Simplifying semi-supervised learning with consistency and confidence., NeurIPS 2020.
>
> [E] Berthelot et al., MixMatch: A Holistic Approach to Semi-Supervised Learning., NeurIPS 2019.
>
> [F] Samuli Laine and Timo Aila. Temporal ensembling for semi-supervised learning. ICLR 2017.
>
> [G] Antti Tarvainen and Harri Valpola. Mean teachers are better role models: Weight-averaged consistency targets improve semi-supervised deep learning results. NeurIPS 2017.
>
> > The statement "Although it achieves effective test-time adaptation, it has a limitation that it can be utilized only if there are BN layers in the trained classifier." is not quite true, because one can always add affine layer anywhere in a pretrained classifier, initialize them as identity and then adapt during TTA.
>
> In Section 3.1., we discuss several options for fine-tuning layers. The statement and the previous one discuss the case when we use the BN layers as fine-tuning layers. Therefore, what Reviewer tB9n suggested (“one can always add affine layer anywhere in a pretrained classifier, initialize them as identity and then adapt during TTA”) could be another option.

---

> > ### Comment · Reviewer_tB9n · 2022-11-18
> > **Feedback**
> >
> > I would like to thank the authors for taking my feedback into account. Some of my concerns have been addressed. Some remaining concerns:
> >  - I find the introduction of an ensemble to compensate an initialization/optimization issue not very convincing. If performance varies strongly between different random initializations, it would be preferable to understand and address the root causes for this rather than building an ensemble, which significantly affects inference time.
> > - In the statement "We observe that TAST can achieve 96% performance gain on average by setting the hyperparameter to (1,1,-1) compared to TAST with the hyperparameter selection", it is not clear to me where I would see a 96% performance gain. Performance rather seems lower. Do the authors mean a drop by 100%-96% = 4%?
> > - "However, when the number of classes is too large, it takes almost all the test time until at least one support example is assigned to all classes. For example, 768 batches out of 782 batches are required on the ImageNet-C experiments until at least one support example is assigned to all classes":  why would it take 768 batches? is this under random shuffling of the test set?
> > - "Since the number of classes (1000) is much larger than the test batch size (64), few prototypes for our method are updated per each test batch while the remaining prototypes remain unupdated. It might affect the performance of the prototype-based classification.": Why is it required to adapt prototypes in every test batch for high performance?
> > - The newly introduced "TAST_BN (w/ fixed prototypes)" seems  a bit pointless since the main point of TAST is to adapt the prototypes or?
> > - TAST-BN is slower than all baselines empirically (Table R1), which limits its utility.

---

> > > ### Author Response · Authors · 2022-11-19
> > > **Response to Feedback by Reviewer tB9n (2/2)**
> > >
> > > > 5. The newly introduced "TAST_BN (w/ fixed prototypes)" seems a bit pointless since the main point of TAST is to adapt the prototypes or?
> > >
> > >
> > > We introduced "TAST_BN (w/ fixed prototypes)" to provide a possible idea of adjusting the original TAST_BN for the case when we encounter too many classes compared to the test batch size. Since we cannot use randomly initialized prototypes for such a case (due to the significant delay in updating all the prototypes during test time), we suggested using the prototypes that are initialized with the weight of the last linear classifier as in TAST. Of course, one can still update the prototypes over the test time, but the performance gain from the updating may not be as significant as before.
> > > Nonetheless, we’d like to point that it is still a meaningful finding that the effective adaptation on ImageNet-C can be achieved with the combination of the prototype approach and self-training (entropy minimization) method.
> > >
> > > > 6. TAST-BN is slower than all baselines empirically (Table R1), which limits its utility.
> > >
> > > We demonstrated that TAST-BN achieves the state-of-the-art performance in image corruption benchmarks, CIFAR-10/100C. Specifically, TAST-BN outperforms TTT++, which is one of state-of-the-art methods, by 1.25% for CIFAR-10C and 4.56% for CIFAR-100C on average, respectively.

---

> > > > ### Comment · Reviewer_tB9n · 2022-11-21
> > > > **Final feedback**
> > > >
> > > > Thanks for clarifying, I have updated my review accordingly.

---

> > > > > ### Author Response · Authors · 2022-11-22
> > > > > **Thanks for the reply**
> > > > >
> > > > > We are glad to hear that our responses have been helpful in understanding our work. We will incorporate the feedback in the next version.
> > > > >
> > > > > Thank you!
> > > > >
> > > > > Authors.

---

> > > ### Author Response · Authors · 2022-11-19
> > > **Response to Feedback by Reviewer tB9n (1/2)**
> > >
> > > Thank you for your interest in our method. We hope that the responses to the questions resolve your concerns.
> > >
> > > > 1. I find the introduction of an ensemble to compensate an initialization/optimization issue not very convincing. If performance varies strongly between different random initializations, it would be preferable to understand and address the root causes for this rather than building an ensemble, which significantly affects inference time.
> > >
> > > We’d like to note that ensemble schemes have been widely used in the existing works [a,b,c] to alleviate the high variance caused by the random initialization.
> > > As we noted in "Runtime comparison" of the common response, TAST requires only ⅓ to ¼ running time compared to the methods that update the entire feature extractors, e.g. SHOT or SHOTIM, even when the number of adaptation modules $N_e$ is set to 20.
> > >
> > > [a] Phung et al., A High-Accuracy Model Average Ensemble of Convolutional Neural Networks for Classification of Cloud Image Patches on Small Datasets, Applied Sciences 2019
> > >
> > > [b] Lee and Chung, Unsupervised Embedding Adaptation via Early-Stage Feature Reconstruction for Few-Shot Classification, ICML 2021
> > >
> > > [c] T. G. Dietterich, Ensemble methods in machine learning, in: International Workshop on Multiple Classifier Systems, Springer, 2000, pp. 1–15.
> > >
> > > > 2. In the statement "We observe that TAST can achieve 96% performance gain on average by setting the hyperparameter to (1,1,-1) compared to TAST with the hyperparameter selection", it is not clear to me where I would see a 96% performance gain. Performance rather seems lower. Do the authors mean a drop by 100%-96% = 4%?
> > >
> > > We reported the relative performance gain of TAST with the fixed hyperparameters (1,1,-1) over that of TAST with the hyperparameter selection method. Specifically, the relative performance gain  is equal to (performance gain of TAST with (1,1,-1))/(performance gain of TAST w/ hyperparameter selection) = 0.96.
> > >
> > > > 3. "However, when the number of classes is too large, it takes almost all the test time until at least one support example is assigned to all classes. For example, 768 batches out of 782 batches are required on the ImageNet-C experiments until at least one support example is assigned to all classes": why would it take 768 batches? is this under random shuffling of the test set?
> > >
> > > The prototype updates are based on the estimated labels of test data by the classifier, not the ground-truth labels. Under test-time domain shift, classifier bias may occur, which may result in assigning most test data only to a subset of classes. Then, even after a large number of batch updates which cover all the ground-truth classes by at least one sample, some prototypes may have not been updated since no previous test data has been classified to those classes. We found that in the above experiments with Gaussian noise, such an event had occurred and it took 768 batches out of 782 batches until all the prototypes were updated at least by once.
> > >
> > > > 4. "Since the number of classes (1000) is much larger than the test batch size (64), few prototypes for our method are updated per each test batch while the remaining prototypes remain unupdated. It might affect the performance of the prototype-based classification.": Why is it required to adapt prototypes in every test batch for high performance?
> > >
> > > The quality of prototypes, i.e., how well each prototype represents the data of the corresponding class in the test domain, is a major factor determining the performance of test-time adaptation. In the online setup, we update the class prototypes by using the classification result of previous test data. To guarantee good performance, it is necessary that all the prototypes are updated and adjusted to the new test domain as quickly as possible over the test time. This is why we prefer adapting as many prototypes as possible in every test batch.
> > > However, on ImageNet-C, too many test batches are required to update all the prototypes since the number of classes (1000) is significantly larger than the test batch size (64).

---

> ### Author Response · Authors · 2022-11-16
> **Response to Reviewer tB9n (2/3)**
>
> > W3-2. In general, TTA methods should have few if any free hyperparameters, so it would be preferable to clearly state default values for hyperparameters and use these everywhere except for ablation studies.
>
> As shown in Table 2 and Figure 2 of the manuscript and Table 8-11 in Appendix C, the extensive ablation studies show that the performance of the adapted classifier is robust to changes in $T$, $\tau$, and $d_\phi$, which are the newly introduced in this work.
> Additionally, we report the experimental results (test accuracy (%)) of TAST with the hyperparameters $(T=1, N_s=1, M=-1, N_e=20)$ on domain generalization benchmarks and compare the results to those of TAST with the given hyperparameter selection method as in T3A.
>
> | Method                                            | VLCS            | PACS              | OfficeHome     | TerraIncognita  | avg           |
> |---------------------------------------------------|-----------------|-------------------|----------------|-----------------|---------------|
> | No Adaptation                                           | 74.88+-0.46  | 79.29+-0.77  | 62.10+-0.31 | 40.62+-1.19  | 64.22         |
> | TAST (T=1, N_s=1, M=-1, N_e=20)                   | 77.04+-0.54    | 81.76+-0.61      | 63.60+-0.52   | 42.81+-0.89    | 66.30 (+2.08) |
> | TAST (w/ hyperparameter selection method, N_e=20) | 77.27+-0.67    | 81.94+-0.44      | 63.70+-0.52   | 42.64+-0.72    | 66.39 (+2.17) |
>
> We observe that TAST can achieve 96% performance gain on average by setting the hyperparameter $(T, N_s, M)$ to (1,1,-1) compared to TAST with the hyperparameter selection. Thus, we can set the default values for hyperparameters for (N_s, T, M, N_e) to (1, 1, -1, 20).
>
> >W4. Experiments on TTA on common corruptions are only conducted on CIFAR10/100-C and not ImageNet-C. Is this because 1000 classes are challenging for TAST?
>
> To obtain good prototypes, a sufficient amount of data per class is required, while we have no access to any labeled data due to TTA settings. Pseudo-labeling alleviates this issue on CIFAR-10/100C, but not on ImageNet-C due to the following two concerns:
> * TAST-BN, which shows the best performance on image corruption benchmarks, requires at least one support example per class to construct prototypes since the support set for TAST-BN is initialized as an empty set. However, when the number of classes is too large, it takes almost all the test time until at least one support example is assigned to all classes. For example, 768 batches out of 782 batches are required on the ImageNet-C experiments until at least one support example is assigned to all classes under Gaussian noise with severity level 5.
> * Since the number of classes (1000) is much larger than the test batch size (64), few prototypes for our method are updated per each test batch while the remaining prototypes remain unupdated. It might affect the performance of the prototype-based classification. To address this issue, it might require a batch size larger than 1000, which is impossible due to the hardware cost.
>
> To alleviate the concerns, we consider a variant of TAST-BN, in which the prototypes are initialized with the weight of the last linear classifier as in TAST and fixed during the test time. We report the experimental results (test accuracy) on ImageNet-C with severity level 5. (when we set $(N_s, M, T)$ to $(1,-1,1)$)
>
> | method  | brit | cont | defoc | elast | fog    | frost  | gauss | glass | impul | jpeg | motn | pixel | shot | snow   | zoom | avg    |
> |---------|------------|----------|--------------|-------------------|--------|--------|----------------|------------|---------------|------------------|-------------|----------|------------|--------|-----------|--------|
> | No Adaptation | 0.5893     | 0.0543   | 0.1792       | 0.1695            | 0.2442 | 0.2331 | 0.0221         | 0.0982     | 0.0185        | 0.3165           | 0.1478      | 0.2061   | 0.0293     | 0.1689 | 0.225     | 0.1801 |
> | TAST_BN (w/ fixed prototypes) | 0.6498     | 0.1926   | 0.167        | 0.4495            | 0.496  | 0.3422 | 0.1665         | 0.1645     | 0.1742        | 0.4183           | 0.2826      | 0.504    | 0.1728     | 0.3615 | 0.4014    | 0.3295 |
>
> We observe that the variant of TAST-BN achieves 14.94% performance gain on average over the corruptions compared to a non-adapted classifier while the prototypes are fixed during the test time.
> We note that test-time adaptations on ImageNet-C are also challenging for TTT++, one of the state-of-the-art test-time methods. https://openreview.net/forum?id=86NHK__yFDl)

---

> ### Author Response · Authors · 2022-11-16
> **Response to Reviewer tB9n (1/3)**
>
> Thank you for the constructive feedback and summary of our work. We hope the responses to the questions and weaknesses below resolve your concerns.
>
> In this work, we propose a novel and unified framework for test-time adaptation by effectively integrating the components (support set utilization, prototype-based classification, adaptation module utilization, and pseudo-label generation), to alleviate the confirmation bias of self-training in test-time adaptation. We demonstrate that our method consistently outperforms the existing test-time adaptation on the standard benchmarks, and all components in our method contribute to the performance gain throughout extensive ablation studies.
> Additionally, we explain the reason for designing our model structure utilizing adaptation modules in “Common Response to Reviewers”. Please see “The ensemble of adaptation modules” in “Common Response to Reviewers”
>
> > W2-1. The ensembling of adaptation modules is not well motivated
>
> Our common response to all reviewers “The motivation of ensemble of adaptation modules” may help address this concern. In our work, we use an ensemble scheme to alleviate the issues caused by the random initialization of adaptation modules. As described in Section 3.1 of the manuscript, we construct a new classifier by adding randomly initialized modules on top of the trained feature extractor. Under the online setting of test-time adaptation, the random initialization of the adaptation module can heavily affect the adaptation. As shown in Table 2 of the manuscript, we found that a single adaptation module can degrade the performance of classifiers. To address this issue, we use multiple adaptation modules to obtain more robust and accurate predictions. Although the performance gain of our method seems to be marginal, our method consistently outperforms the current state-of-the-art test-time adaptation methods on two standard benchmarks, domain generalization and image corruption.
>
> > W2-2. if one employs an ensemble, wouldn't it be natural/preferable to also employ different support sets in conjunction with different adaptation models in order to increase diversity within the ensemble?
>
> Employing a different support set for each adaptation module using its own outputs may incur considerable bias or inefficiency due to the random initialization of the modules. TAST builds prototypes using the support set and adapts the classifiers to match the prototype-based label distribution and the neighborhood label distribution of the test data. If both the support set and prototype constructions are affected by the adaptation module outputs, then the learning of the adaptation module at the beginning of test time may be more affected by the random initialization of the modules. To address this issue, we construct the support set as described in Eq 2-3 and the modules share the same support set.
>
> > W3-1. The method TAST has several additional hyperparameters (N_s, T, M, N_e) and it is not entirely clear how these have been chosen in certain experiments. In particular, they should not be chosen in any way based on the target domains.
>
> As described in Section 4.1.1 of the manuscript and Appendix A.1, we follow the hyperparameter selection method used in T3A. We split each dataset of training domains into 80% and 20% and use the smaller set as the validation set. We use the validation set to select hyperparameters. Thus, we do not use target domain information to select hyperparameters.

---

### Official Review · Reviewer_Lha6 · 2022-10-24

**Confidence:** 4
**Correctness:** 4
**Technical Novelty And Significance:** 3
**Empirical Novelty And Significance:** 3
**Recommendation:** 8

**Clarity, Quality, Novelty And Reproducibility:**

This paper was well-written and organized. The proposed technique was clearly described and referred to all the papers where it was needed. The novelty lies in using the previous data efficiently during test time such that the unreliable samples were discarded and only the confident examples made it to the support set.

**Details Of Ethics Concerns:**

None.

**Strength And Weaknesses:**

Strength:
+The proposed method's main strength is introducing adaptation module(s) that were randomly initialized and fine-tuned using the nearest neighbor samples from test data. The author also introduced TAST-BN, which fine-tunes the BN layer (if it exists) instead of the adaptation module.
+ This method outperformed other techniques in image corruption datasets. The empirical results show the benefit of the method, and some ablation studies are provided.
+The supplementary material provides additional implementation details and experimental results that help support the paper.
+ Also, since the codes are also provided in supplementary material, there is less concern that the results in this paper would be difficult for a reader to reproduce.  The paper includes information that would make it possible to reproduce the methods and experiments (but not with MIDRC data).


Weaknesses:
+ The proposed method is computationally expensive. Although, the author compares their method with others in terms of computation in an ablation study, and mentions that the proposed method (TAST) was computationally expensive. Still, their method takes 4x more time than the state-of-the-art (T3A). This performance is not ideal, especially when adapting the model on test time.

**Summary Of The Paper:**

The author proposed a test-time adaptation via a self-training method (TAST). Introducing an adaptation module to generate pseudo-labels through the nearest neighbor approach. Based on the T3A method, the author incorporated a support set created using previous data and a prediction from a classifier. The support set only contains examples with a reliable pseudo label, which is determined by calculating the cosine similarity between the test sample x and its nearest neighbor in the embedding space. The multiple adaptation module is randomly initialized, trained during the test time, and then ensemble the result generated from different adaptation modules to predict the labels for test data. The method outperforms state-of-art results on domain generalization on corrupted image corruption datasets.

**Summary Of The Review:**

This work delivers on its claimed contributions, although the time complexity may limit it application at test time. I anticipate it will be a valuable publication for researchers in test time adaptation.

---

> ### Author Response · Authors · 2022-11-16
> **Response to Reviewer Lha6**
>
> Thank you for the positive and thorough review and for acknowledging the sufficient novelty of our work.
>
> > W1. The proposed method is computationally expensive. Although, the author compares their method with others in terms of computation in an ablation study, and mentions that the proposed method (TAST) was computationally expensive. Still, their method takes 4x more time than the state-of-the-art (T3A). This performance is not ideal, especially when adapting the model on test time.
>
> We reported an average runtime spent to adapt classifiers during test time in Table 4 of Appendix A. We additionally report (1) the number of learnable parameters during adaptation and (2) an average runtime to adapt trained classifiers with a single hyperparameter combination for all test-time adaptation algorithms in Table R1-2 of “Common Response to Reviewers”.

---

### Official Review · Reviewer_rMac · 2022-10-24

**Confidence:** 3
**Correctness:** 3
**Technical Novelty And Significance:** 2
**Empirical Novelty And Significance:** 2
**Recommendation:** 5

**Clarity, Quality, Novelty And Reproducibility:**

The paper is generally clearly written and has novelty for the specific application of TTA. It seems reproducible and hopefully authors will share code

**Strength And Weaknesses:**

### Strengths

S1: The method uses "pseudo-label distributions for test data using the nearest neighbor information" an interesting idea and novel in a test-time adaptation setting

S2: The method improves results on different benchmarks

### Weaknesses

W1: **Transductive vs non-transducive test setups mixed**. A support set is needed and kept updated at test time  to define (pseudo)-prototypes for the proposed method, similar to T3A but unlike other methods like TENT . This seems like a strong enough assumption since now one could call the test setup "transductive" and authors should explicitly mention this eg in Table 1, and separate the methods accordingly.

W2: **Differences with related works are conceptually small**. The proposed approach adds adaptors a projector for TTA, something generally common in many other settings (see N1) and instead of constructing prototypes as in T3A, it uses a neiarest neighbor classifiers. TAST-BN is very similar to other TTA methods that only updated BN parameters. It is unclear what the intuitions behind this specific design is and what insights it offers.


W4: **discussion on efficiency missing**.How much is the added compute and parameters vs other methods?

### Notes and questions

Q1: **architecture of adaptors**. What is the architecture of h? this should be more prominently featured

N1: **Discussion on the use of projectors beside TTA** is missing. From SSL to SL projectors are being used during training o help transfer performance. SSL methods like AsimSiam, SImSiam, SimCLR and many others, as well as supervised methods like SL-MLP (Revisiting the Transferability of Supervised Pretraining: an MLP Perspective, CVPR 2022) or t-ReX (Improving the Generalization of Supervised Models, arxiv 2022).

N2: **Missing refs utilizing nearest neighbor information during training**. Some missing references that learn using nearest neighbor information in similar ways and should be discussed/cited - see below. Moreover, "self-training with nearest neighbor information" is how most manifold learning methods operate -  some discussion on that should be added.

- Iscen, Ahmet, et al. "Learning with Neighbor Consistency for Noisy Labels." CVPR 2022.

- Almazán, Jon, et al. "Granularity-aware Adaptation for Image Retrieval over Multiple Tasks." ECCV 2022

Note that although for a different setting, Almazán, Jon, et al. also use multiple adaptors that are trained using pseudolabels, while also fused using nearest neighbor-based pseudo labels or by averaging the multiple adaptors



**Summary Of The Paper:**

The paper proposes a test time adaptation (TTA) method named TAST where a trainable projector is added at test time on top of the feature extractor and optimised via information from the top nearest neighbors.

**Summary Of The Review:**

Although an interesting method with some technical novelty, intuitions on the design choices and motivation are missing, as well as discussions/related work of the use of the same techniques outside TTA. I encourage the authors to respond to the concerns above and I am willing to increase my score accordingly with clear and satisfying answers.

---

> ### Author Response · Authors · 2022-11-16
> **Response to Reviewer rMac (2/2)**
>
> >W4: discussion on efficiency missing. How much is the added compute and parameters vs other methods?
>
> We reported an average runtime spent to adapt classifiers during test time in Table 4 of Appendix A. We additionally report (1) the number of learnable parameters during adaptation and (2) an average runtime to adapt trained classifiers with a single hyperparameter combination for all test-time adaptation algorithms in Table R1-2 of “Common Response to Reviewers”.
>
> > N1: Discussion on the use of projectors beside TTA is missing.
>
> The adaptation module structure is used in many fields such as self-supervised learning (which is called “projection head”) as Reviewer rMac informed. Although the existing methods mainly focus on training time, TAST focuses on test time. For example, SimCLR adds a projection head on the top of a feature extractor at the beginning of training time and trains the feature extractor and the projection head with an instance discrimination loss. After the training time, for downstream tasks, SimCLR uses feature extractor outputs rather than projection head ones. However, TAST adds adaptation modules at the beginning of test time and trains the modules with the nearest neighbor-based pseudo-label distribution. To predict the label of test data, we use the averaged predicted class distribution from the adaptation modules.
>
> > N2: Missing refs utilizing nearest neighbor information during training.
>
> Thank the reviewer for sharing [A] and [B]. Like [A], there are so many studies using the nearest neighbor information in various fields including metric learning and few-shot learning. Thus, in Section 5 of the manuscript, we summarize the existing works using the nearest neighbor information in source-free domain adaptation, which deals with a test setup similar to that of test-time adaptation but focuses on offline instead of online.
> As Reviewer rMac mentioned, [B] also uses nearest neighbor information and multiple adaptation modules. However, there are two main differences between TAST and [D].
> * (Pseudo-labeling method) [B] generates pseudo-labels of unlabeled data using k-means. [B] obtains a set of different pseudo-labels by using k-means with different k. However, TAST defines a pseudo-label distribution for the unlabeled data considering nearest neighbor information with a support set that is composed of previous test data and their predictions.
> * (Purpose of utilizing multiple adaptation modules) In [B], each adaptation module is trained with the different pseudo labels individually. Thus, the number of adaptation modules is the same as the number of the generated pseudo labels. However, as described in “The ensemble of adaptation modules” of “Common Response to Reviewers”, we use the ensemble of adaptation modules to mitigate the performance degradation caused by the random initialization of the modules. Thus, all adaptation modules share the same pseudo-label distributions, and the number of adaptation modules is not fixed.
>
> [A] Iscen, Ahmet, et al. "Learning with Neighbor Consistency for Noisy Labels." CVPR 2022.
> [B] Almazán, Jon, et al. "Granularity-aware Adaptation for Image Retrieval over Multiple Tasks." ECCV 2022

---

> > ### Comment · Reviewer_rMac · 2022-11-23
> > **Thank you for the response**
> >
> > (note that this comment was first posted on 19 Nov 2022, but I accidentally posted it uder the wrong review)
> >
> > I thank the reviewers for the detailed response. I also thank them for taking into account the feedback and updating Table 1 (T1) and for showing extensive runtime experiments. The updated manuscript is more complete than the previous one.
> >
> > However, the clarity that the new T1 and T4 bring, raises an important concern: Having a "memory" (i.e. the "transductive" test setting) brings gains in performance, and the authors only compare to one other such method, T3A. Performance between the two is more or less the same (taking std dev into account) while T3A is orders of magnitude faster.
> >
> > I will therefore leave my score as borderline reject.

---

> ### Author Response · Authors · 2022-11-16
> **Response to Reviewer rMac (1/2)**
>
> Thank you for the constructive feedback and for acknowledging our experimental analysis. We hope the responses to the notes, questions, and weaknesses below resolve your concerns.
> > Summary. Although an interesting method with some technical novelty, intuitions on the design choices and motivation are missing, as well as discussions/related work of the use of the same techniques outside TTA.
>
> In this work, we propose a novel and unified framework for test-time adaptation by effectively integrating the components (support set utilization, prototype-based classification, adaptation module utilization, and pseudo-label generation), to alleviate the confirmation bias of self-training in test-time adaptation. We demonstrate that our method consistently outperforms the existing test-time adaptation on the standard benchmarks, and all components in our method contribute to the performance gain throughout extensive ablation studies.
> Additionally, we explain the reason for designing our model structure utilizing adaptation modules in “Common Response to Reviewers”. Please see “The ensemble of adaptation modules” in “Common Response to Reviewers”
>
> > Q1. architecture of adaptors. What is the architecture of h? this should be more prominently featured
>
> As described in line 2 of page 7 in the manuscript, we utilize BatchEnsemble (BE) for the adaptation modules of TAST. BE is a simple and efficient ensemble method that greatly reduces the computational cost by weight-sharing. Each ensemble member of BE is composed of two layers with a shared weight and rank-one factors. Specifically, the weight matrix of j-th ensemble member is $W \circ r_j s_j^T$ where $W$ is a shared weight and $r_j s_j^T$ is the rank-one factor of j-th ensemble member. Although the existing deep ensemble (DE) methods do not share any weights, all ensemble members share W, and thus BE reduces the number of parameters compared to DE. Moreover, unlike DE, only the last layer of all ensemble members of BE are different, and thus it can be easily vectorized and trained simultaneously. Therefore, BE greatly reduces the computation cost. We add this detailed explanation about BE in Appendix A.
>
> > W1. Transductive vs non-transducive test setups mixed.
>
> We modified Table 1 to inform which methods use a memory, storing (a subset of) previous test data and their predictions, during test time in Table 1 of the manuscript. We would like to mention that the amount of required memory is not significant for TAST and TAST-BN though since TAST stores light-weight feature embeddings instead of test images, and TAST-BN restricts the support set size while TAST-BN stores the test images itself, as described in Section 3.2 and Appendix A.
>
> > W2: Differences with related works are conceptually small.
>
> As Reviewer hpzb pointed out, the proposed TAST is a novel assembly of existing choices for test-time adaptation, which in particular draws from entropy minimization methods and prototype methods to define a new hybrid adaptation technique that uses both. TAST is the first method that combines adaptation by exemplar (neighbor), prototype (cluster), and entropy minimization (optimizing). By experiments, we demonstrated that this novel assembly of existing methods effectively alleviates the confirmation bias of self-training in test-time adaptation. In particular, by the ablation study summarized in Table 2-3 and Fig 1 of the manuscript, we showed that all components of TAST contribute to the performance gain.

---

### Official Review · Reviewer_hpzb · 2022-10-24

**Confidence:** 5
**Correctness:** 4
**Technical Novelty And Significance:** 2
**Empirical Novelty And Significance:** 2
**Recommendation:** 6

**Clarity, Quality, Novelty And Reproducibility:**

Clarity: The clarity is good. The main idea and details of the method are communicated by the text, figures (especially Figure 1), and pseudo-code in Algorithm 1, which provides a variety of formats for comprehension. The notation is clear and consistent throughout, with the right amount to specify the setting and method without obscuring them behind too many symbols.The organization into sections and subsections is sensible and easy to navigate. While there are a few minor points to improve clarity (see Misc. Feedback below), they can all be resolved and do not harm understanding.

Quality: The baselines and comparisons are broad, appropriate, and representative of recent and strong test-time adaptation methods. The experiments cover not only prior methods exactly as they were proposed, but also include variations that are relevant to the parameterization of the proposed TAST, for instance TentBN which adds new parameters and TentClf which only trains the last parameters. While the experimental design is good, the significance of the results is minor, as the combined total effect of the nearest neighbor information and new adaptation layers is barely an improvement.

Novelty: The proposed TAST is a novel assembly of existing choices for test-time adaptation, which in particular draws from entropy minimization methods and prototype methods to define a new hybrid adaptation technique that uses both. The use of prototypes to define targets follows SHOT and T3A, but this work also includes nearest neighbors of target points in the assignment of pseudo-labels. The use of multiple adapter layers as an ensemble is not new to TTA, as that that was done by BACS (Zhou & Levine, NeurIPS'21), though the test-time ensemble parameters differ. There is empirical novelty in comparing more TTA methods and TTA version of SFDA methods (SHOT) side-by-side on domain generalization benchmarks. For more related work on mixed prototype/exemplar inference, the work could consider citing and discussing IMP (Allen et al., ICML'19) and the multi-prototype representation of Mensink et al. PAMI'13.

Reproducibility: This work could be reproduced, as the exposition is sufficiently detailed, and the method itself is not too intricate. The method could also be completely specified by releasing the code, but the submission does not indicate whether or not this will be done.

questions:

- What is the computational cost of TAST compared to T3A and compared to Tent? This could be a wallclock measurement, or an abstract measurement such as the number of forwards and gradients in the network.

miscellaneous feedback:

- Abstract:
  - Please consider mentioning the datasets along with the tasks to know to highlight the experimental scope early on.
- Introduction:
  - "two popular categories for TTA" consider including a third category for normalization or statistics-based methods, such as only updating batch normalization statistics (see Scheider et al. NeurIPS'20).
- Table 1: Consider marking the best non-TAST result, the prior state-of-the-art, by underlining to make it easier to compare against TAST. As an aside, it is striking that so many test-time adaptation methods harm accuracy on domain generalization benchmarks. This is worth noting in the caption or the corresponding text of Section 4.1.
- Figure 2: please remind the reader in the caption or text of the meaning of each hyperparameter, because the notation introduced by Algorithm 1 is pages away.
- Notation: NN could be confused as either "nearest neighbor" or "neural network" so consider "N(x; S)" as a notation for the "neighborhood" of x in S instead.
- Method name: TAST for "Test-time Adaptive Self Training" is too generic, because it could describe any number of existing methods like TENT, MEMO, BACS, etc. The nearest neighbor part of the proposed method is more unique, so could an ancronym including "N" for "neighbor" be more distinctive?
- Other naming: "TentBN" is confusing at first, because Tent already adapts normalization layer. Consider "TentAdapter" or "TentFinal" to designate Tent with a last, new adapter layer included.
- Proofreading:
  - "Especially, we use [...]" after Eq. 5 should be "Specifically, we use [...]"


**Strength And Weaknesses:**

*strengths*

- TAST only updates parameters of the deepest, final layers so it does not need to backpropagate gradients through the entire model, unlike methods like TENT/MEMO/SHOT that update parameters in all normalization layers/all layers/all but the last layer, respectively. This could improve computational efficiency, if it is not cancelled out by the number of new final layers that TAST adds and the number of updates needed to train them.
- The domain generalization datasets diversely include images that are real (VLCS, OfficeHome), stylized/synthetic (PACS), and scientific (TerraIncognita) and follow the established choices of prior work on test-time adaptation (T3A). The ablation of including nearest neighbor predictions without updates ("Effect of nearest neighbor information") does show a slight (<1%) effect of of neighbors alone, which validates the idea of the proposed method.
- The image corruption benchmarks are standard, but small. CIFAR-10/100-C are common, but ImageNet-C is preferred, as it is more representative of the scale of current image classifiers. prior work such as Tent and SHOT report on ImageNet-C for this reason.
- The results are insensitive to batch size across 8-128 and show improvement over ERM at each size. However, the minimum viable batch size is not reported, which would be worth knowing to determine which methods are competitive at small batch sizes and in the limit batch size one.

*weaknesses*

- The effect of TAST is small. generally improves by 1 point or less, with the exception of PACS with +6 points (Table 1), and this gap is only achieved with a higher number of adapters. The ablation of $N_e$ for the number of adapters shows that the improvement with no adapters is only recovered at 5 adapters, which has greater computational cost than none.
- More memory and computation is needed to maintain a set of neighbors, as done by TAST, than to maintain a set of prototypes, as done by T3A.
- The results are sensitive to the choice of parameters to update, and which choice is best varies across datasets. In particular neither TAST, with updates to new modules, nor TAST-BN, with updates to normalization layers, is a clear winner. Without a model selection rule to choose between them it is not clear which to apply.
- The method has hyperparameters to determine the threshold between reliable and unreliable pseudo-labels (eq. 2), like T3A, but there are no ablations to check the sensitivity of results to this reliability rule. If it needs tuning then it would limit the general applicability of the method.
- There is closely related work on source-free domain adaptation with neighbors, which is cited (Tang et a. 2021 and Yang et al. 2021), but these works do nevertheless reduce the novelty of the proposed TAST.


**Summary Of The Paper:**

In the setting of test-time adaptation, an adaptation method tries to update a model given only online access to test data in order to reduce generalization error without altering the training process. The proposed test-time adaptation via self-training with nearest neighbors method (TAST), differs from existing methods in (1) its choice of parameters for adaptation, (2) its use of nearest neighbors in defining the targets for self-training updates, and (3) it's combination of prototype (average) and exemplar (neighbor) predictions. The parameters for adaptation are those of newly-initialized predictors on top of the trained feature extractor, including multiple predictors acting as an ensemble, in order to provide a variety of predictions to average over for robustness. TAST differs from prior prototypical methods like T3A by making use of neighbors to update the representation, while T3A keeps the representation fixed which limits the range of its improvement. Experiments on standard benchmarks for domain generalization (VLCS, PACS, OfficeHome, TerraIncognita) and image corruption (CIFAR-10/100-C) show marginal improvements of +1 point of accuracy (with the exception of a +6 point gain on PACS). This marginal improvement is above the state-of-the-art methods for test-time adaptation including entropy minimization and prototypical approaches.

**Summary Of The Review:**

TAST is a technically correct and slightly novel test-time adaptation method that combines adaptation by exemplar (neighbor), prototype (cluster), and entropy minimization (optimizing). It achieves a slight improvement in accuracy (around 1 percentage point) for multiple domain generalization and image corruption datasets, but only reaches this improvement by requiring more computation time and memory for inference and adaptation. As such it is a borderline paper, but it can inform the community about the possibility of improving test-time adaptation results by combining clustering based on neighbors with entropy minimization of newly-included adaptation layers without updating the parameters from training.

---

> ### Author Response · Authors · 2022-11-16
> **Response to Reviewer hpzb (3/3)**
>
> > Method name: TAST for "Test-time Adaptive Self Training" is too generic, because it could describe any number of existing methods like TENT, MEMO, BACS, etc. The nearest neighbor part of the proposed method is more unique, so could an ancronym including "N" for "neighbor" be more distinctive?
>
> As we confirmed, we cannot currently modify the title and abstract on OpenReview. Thus, we will change the method name together with the title and abstract in the final version.
>
> > S3. The image corruption benchmarks are standard, but small. CIFAR-10/100-C are common, but ImageNet-C is preferred, as it is more representative of the scale of current image classifiers. prior work such as Tent and SHOT report on ImageNet-C for this reason.
>
> To obtain good prototypes, a sufficient amount of data per class is required, while we have no access to any labeled data due to TTA settings. Pseudo-labeling alleviates this issue on CIFAR-10/100C, but not on ImageNet-C due to the following two concerns:
> * TAST-BN, which shows the best performance on image corruption benchmarks, requires at least one support example per class to construct prototypes since the support set for TAST-BN is initialized as an empty set. However, when the number of classes is too large, it takes almost all the test time until at least one support example is assigned to all classes. For example, 768 batches out of 782 batches are required on the ImageNet-C experiments until at least one support example is assigned to all classes under Gaussian noise with severity level 5.
> * Since the number of classes (1000) is much larger than the test batch size (64), few prototypes for our method are updated per each test batch while the remaining prototypes remain unupdated. It might affect the performance of the prototype-based classification. To address this issue, it might require a batch size larger than 1000, which is impossible due to the hardware cost.
>
> To alleviate the concerns, we consider a variant of TAST-BN, in which the prototypes are initialized with the weight of the last linear classifier as in TAST and fixed during the test time. We report the experimental results (test accuracy) on ImageNet-C with severity level 5. (when we set $(N_s, M, T)$ to $(1,-1,1)$)
>
> | method  | brit | cont | defoc | elast | fog    | frost  | gauss | glass | impul | jpeg | motn | pixel | shot | snow   | zoom | avg    |
> |---------|------------|----------|--------------|-------------------|--------|--------|----------------|------------|---------------|------------------|-------------|----------|------------|--------|-----------|--------|
> | No Adaptation | 0.5893     | 0.0543   | 0.1792       | 0.1695            | 0.2442 | 0.2331 | 0.0221         | 0.0982     | 0.0185        | 0.3165           | 0.1478      | 0.2061   | 0.0293     | 0.1689 | 0.225     | 0.1801 |
> | TAST_BN (w/ fixed prototypes) | 0.6498     | 0.1926   | 0.167        | 0.4495            | 0.496  | 0.3422 | 0.1665         | 0.1645     | 0.1742        | 0.4183           | 0.2826      | 0.504    | 0.1728     | 0.3615 | 0.4014    | 0.3295 |
>
> We observe that the variant of TAST-BN achieves 14.94% performance gain on average over the corruptions compared to a non-adapted classifier while the prototypes are fixed during the test time.
> We note that test-time adaptations on ImageNet-C are also challenging for TTT++, one of the state-of-the-art test-time methods. https://openreview.net/forum?id=86NHK__yFDl)

---

> > ### Comment · Reviewer_hpzb · 2022-11-27
> > **Thank you for the thorough response.**
> >
> > Thank you for the reply to specific points in this review and for the general response to all reviewers. I maintain my recommendation of borderline acceptance, given the clarification and further results, because some are positive and some are negative. On the whole I still side with acceptance because this work informs the comparing and contrasting of optimization (entropy minimization) and clustering (prototypical) methods.
> >
> > Positives:
> >
> > - Runtime: The timings measured in the response are informative. This table make interesting points, about the balance of computation for gradients vs. neighbors, and about the effect of how deep the parameters are in the network. It should be included in the paper or appendix.
> > - Reproducibility: This issue is resolved, as the code is in the supplement, and more details have been provided in the revision.
> > - Hyperparameter Sensitivity: The response shares an experiment that resolves the concern about over-sensitivity to M. While TAST still has a number of hyperparameters compared to other methods, it does not seem to need an excessive amount of tuning. The points made in the response about domain generalization vs. corruptions are fair, and needing a different parameterization for each can highlight future work for a unified test-time adaptation method.
> >
> > Negatives:
> >
> > - The tradeoff between updating features vs. prototypes is not described well. This has come up during the discussion, in particular for its impact in the larger-scale/many-class benchmark of ImageNet-C, but it is not discussed in the main paper. However, this could be a contribution of the paper if it were better discussed. Please see the similar comments by Reviewer tB9n.
> > - Related Work: BACS is only in the Appendix A.5, rather than the related work or introduction of the main paper, even though it is an ensembling test-time training method. It should be considered for mention in the main paper, because the distinction between optimizing an ensemble for adaptation during training (BACS) and instantiating an ensemble for adaptation during testing (this work: TAST) can help orient the reader to train-time vs. test-time adaptation.
> > - The reporting of results for test-time adaptation to corruption is not clear and balanced. TAST is not competitive on ImageNet-C, so its state-of-the-art claim for corruptions should be restricted to small-scale datasets (CIFAR-10/100-C).
> > - Several results, including the result on ImageNet-C in the response, sets M = -1 to keep _all_ test-time examples. This memory that is linear in the size of test set is nothing like the constant memory usage of methods that make gradient updates (Tent) or prototypes with a fixed support set size or the possibility of incremental computation (T3A).

---

> > > ### Author Response · Authors · 2022-11-29
> > > **Thanks for the reply**
> > >
> > > Thank you for the positive comments and the suggestion to better explain our work. We will modify it in the final version.
> > > Additionally, we hope that the additional responses resolve your concerns.
> > >
> > > > N3. The reporting of results for test-time adaptation to corruption is not clear and balanced. TAST is not competitive on ImageNet-C, so its state-of-the-art claim for corruptions should be restricted to small-scale datasets (CIFAR-10/100-C).
> > >
> > > To make it clear, we identified the benchmarks in which our method achieved state-of-the-art performance in the abstract of the revised version. Nonetheless, on ImageNet-C experiments, we’d like to point out that it is still a meaningful finding that the effective adaptation can be achieved with the combination of the prototype approach and self-training (entropy minimization) method.
> > >
> > > > N4. Several results, including the result on ImageNet-C in the response, sets M = -1 to keep all test-time examples. This memory that is linear in the size of test set is nothing like the constant memory usage of methods that make gradient updates (Tent) or prototypes with a fixed support set size or the possibility of incremental computation (T3A).
> > >
> > > The experiments with $M=-1$ were not firstly used in our method, but in T3A.
> > > As described in Section 4.1.1 of the manuscript, we use the same set of possible values for each hyperparameter with the existing methods. For example, TAST and T3A use the same set of possible values for $M$ ($M \in$ {1,5,20,50,100,-1}).

---

> ### Author Response · Authors · 2022-11-16
> **Response to Reviewer hpzb (2/3)**
>
> > W3. The results are sensitive to the choice of parameters to update, and which choice is best varies across datasets. In particular neither TAST, with updates to new modules, nor TAST-BN, with updates to normalization layers, is a clear winner. Without a model selection rule to choose between them it is not clear which to apply.
>
> We observe that the best test-time adaptation method which achieves effective adaptation in each benchmark can be different in general since the benchmarks deal with very different types of domain/distribution shifts. For example, in domain generalization benchmarks, the styles of images between the training and test domain are significantly different. On image corruption benchmarks, on the other hand, training and test datasets consist of the same-style but clean/corrupted images, respectively. From the results of two benchmarks (Table 1 and 3), we could observe some common trends that can be summarized as below.
> * On the image corruption benchmarks, the test-time adaptation algorithms using the frozen feature extractor such as T3A and TAST showed poor performance compared to those using the adapted feature extractor including Tent and TAST-BN.
> * On the domain generalization benchmarks, we observe that the performance of most baseline methods, which fine-tune the feature extractors, is lower than that of the classifier without adaptation.
>
> We can understand that CIFAR-10/100C require adapting the entire feature extractor, and thus only TAST-BN can be a competitive method here, while updating the entire feature extractor is not effective in domain generalization benchmarks. Even though the adaptation modules are different between TAST and TAST-BN, the two methods share the common idea of improving the test-time adaptation by combining clustering based on neighbors with entropy minimization of adaptation layers. We demonstrated the effectiveness of this methodology over the two standard benchmarks, even though the winner between TAST and TAST-BN can be different depending on the benchmarks.
>
> > W4.  The method has hyperparameters to determine the threshold between reliable and unreliable pseudo-labels (eq. 2), like T3A, but there are no ablations to check the sensitivity of results to this reliability rule. If it needs tuning then it would limit the general applicability of the method.
>
> We conducted experiments to check the sensitivity of TAST over M, the number of support examples per class, and the results are summarized in Table R3. We can observe that TAST is robust against the change in $M$ and entropy-based filtering to obtain reliable pseudo labels is effective.
>
> **R3. Sensitivity analysis about the number of support examples per class $M$. Average accuracy on test environment A using classifiers learned by ERM on PACS**
> |             |        |        | M       |        |        |        |
> |-------------|--------|--------|--------|--------|--------|--------|
> |             | 1      | 5      | 20     | 50     | 100    | -1     |
> | TAST (ours) | 0.7822 | 0.7868 | 0.7953 | 0.8001 | 0.8030 | 0.7998 |
>
> > W5.  There is closely related work on source-free domain adaptation with neighbors, which is cited (Tang et a. 2021 and Yang et al. 2021), but these works do nevertheless reduce the novelty of the proposed TAST.
>
> We have summarized the novelty of our method.
> * We propose a novel and unified framework by effectively integrating the components (support set utilization, prototype-based classification, adaptation module utilization, and pseudo-label generation), to alleviate the confirmation bias of self-training in test-time adaptation.
> * We demonstrate that TAST outperforms the current state-of-the-art test-time adaptation algorithms on two standard settings, domain generalization and image corruption. Extensive ablation studies show that both the pseudo-label distribution considering nearest neighbor information and the adaptation module utilization contribute to the performance increase.

---

> ### Author Response · Authors · 2022-11-16
> **Response to Reviewer hpzb (1/3)**
>
> Thank you for the positive feedback and summary of our work. We hope the responses to the questions and weaknesses below resolve your concerns
>
> >Q1/W2. Computational cost What is the computational cost of TAST compared to T3A and compared to Tent? This could be a wallclock measurement, or an abstract measurement such as the number of forwards and gradients in the network.
>
> We reported an average runtime spent to adapt classifiers during test time in Table 4 of Appendix A. We additionally report an average runtime to adapt trained classifiers with a single hyperparameter combination for all test-time adaptation algorithms in Table R1 of “Common Response to Reviewers”.
>
>
> > W1-1. The effect of TAST is small. generally improves by 1 point or less, with the exception of PACS with +6 points (Table 1), and this gap is only achieved with a higher number of adapters.
>
> We note that, as shown in Table 1 of the manuscript, TAST/TAST-BN consistently show the best performance on the domain generalization benchmarks, although the best performer among the existing methods is different for each benchmark.
>
> > W1-2. The ablation of $N_e$ for the number of adapters shows that the improvement with no adapters is only recovered at 5 adapters, which has greater computational cost than none.
>
> Our common response to all reviewers “The motivation of ensemble of adaptation modules” may help address this concern. In our work, we use an ensemble scheme to alleviate the issues caused by the random initialization of adaptation modules. As described in Section 3.1 of the manuscript, we construct a new classifier by adding randomly initialized modules on top of the trained feature extractor. Under the online setting of test-time adaptation, the random initialization of the adaptation module can heavily affect the adaptation. As shown in Table 2 of the manuscript, we found that a single adaptation module can degrade the performance of classifiers. To address this issue, we use multiple adaptation modules to obtain more robust and accurate predictions. Although the performance gain of our method seems to be marginal, our method consistently outperforms the current state-of-the-art test-time adaptation methods on two standard benchmarks, domain generalization and image corruption.

---

### Author Response · Authors · 2022-11-16
**Common response to reviewers**

We thank the reviewers for their time and efforts to provide constructive comments. We summarize the common concerns and questions raised by the reviewers.
* Runtime comparison (Reviewer hpzb, Lha6, tB9n)
* The motivation of adaptation modules (Reviewer rMac, tB9n)
* Reproducibility (Reviewer hpzb, rMac, tB9n)

The detailed responses and answers to other concerns/questions are written in separate comments. We believe that we successfully address the concerns and questions raised by the reviewers. If you have any other concerns or questions, please post a comment at any moment for us to respond.

### **1. Runtime comparison**

In Table 4 of Appendix A, we reported the total running time for each test-time adaptation algorithm. The reported time includes the time to conduct experiments for hyperparameter search for all the considered combinations of hyperparameter choices. Since TAST and TAST-BN have more hyperparameters compared to other algorithms, they require longer running time.

For a fair comparison, we also report the average runtime to adapt trained classifiers with a single hyperparameter combination (whenever possible we set $(N_e, T, N_s, M)$ to (20,1,8,-1)) for all test-time adaptation algorithms in Table R1. We note that TAST, which updates the support set and the adaptation modules, requires only ⅓ to ¼ running time compared to the methods that update the entire feature extractors, e.g. SHOT or SHOTIM. On the other hand, TAST-BN, which updates the support set as well as the BN layer, requires more running time (about 2x) compared to SHOT or SHOTIM. The overhead is not significant though due to the online setting.

**Table R1. Running time (sec) to adapt classifiers that use ResNet-18 as a backbone network with a single hyperparameter combination $(N_e=20, T=1, N_s=8, M=-1)$**

| Method         | VLCS   | PACS   | OfficeHome | TerraIncognita  |
|----------------|--------|--------|------------|-----------------|
| Tent           | 53.32  | 15.17  | 50.56      | 76.48           |
| TentAdapter    | 0.59   | 0.43   | 0.64       | 0.92            |
| TentClf        | 0.52   | 0.40   | 0.60       | 0.81            |
| SHOT           | 55.10  | 20.97  | 54.66      | 77.08           |
| SHOTIM         | 54.82  | 20.73  | 54.63      | 77.00           |
| PL             | 55.03  | 20.75  | 54.53      | 77.02           |
| PLClf          | 0.57   | 0.43   | 0.62       | 0.92            |
| T3A            | 0.62   | 0.58   | 3.44       | 1.61            |
| TAST (Ours)    | 7.71   | 6.92   | 12.74      | 23.69           |
| TAST-BN (Ours) | 81.54  | 73.93  | 114.33     | 179.48          |

We further report the number of learnable parameters during adaptation for each test-time adaptation method in Table R2. The number of learnable parameters of TAST is not significantly large compared to those of existing test-time adaptation methods, even though the number of adaptation modules $N_e$ is set to 20.

**Table R2. A number of learnable parameters using ResNet-18 backbones on PACS**
| Method          | # learnable parameters      |
|-----------------|-----------------------------|
| Tent            | 13191                       |
| TentAdapter     | 14215                       |
| TentClf         | 3591                        |
| SHOT            | 11176512                    |
| SHOTIM          | 11176512                    |
| PL              | 11180103                    |
| PLClf           | 3591                        |
| TAST (ours)     | 80896 (when $N_e$=20 )        |
| TAST-BN (ours)  | 13191                       |

### **2. The motivation of ensemble of adaptation modules**

In our work, we use an ensemble scheme to alleviate the issues caused by the random initialization of adaptation modules. As described in Section 3.1 of the manuscript, we construct a new classifier by adding randomly initialized modules on top of the trained feature extractor. Under the online setting of test-time adaptation, the random initialization of the adaptation module can heavily affect the adaptation. As shown in Table 2 of the manuscript, we found that a single adaptation module can degrade the performance of classifiers. To address this issue, we use multiple adaptation modules to obtain more robust and accurate predictions. Although the performance gain of our method seems to be marginal, our method consistently outperforms the current state-of-the-art test-time adaptation methods on two standard benchmarks, domain generalization and image corruption.

### **3. Reproducibility**

The implementation codes were provided in the **supplementary file**, as reviewer Lha6 commented. And, we correct the abstract to indicate that the implementation is available in the supplementary file.

---

### Author Response · Authors · 2022-11-16
**The revised paper is uploaded**

We have uploaded our revised paper with the following modification:
* (Section 3.1) We modify the motivation of ensemble of adaptation modules.
* (Appendix A.3) We add more details on the adaptation module structure.
* (Table 4) We summarize the average runtime to adapt trained classifiers with a single hyperparameter combination instead of reporting the total running time including hyperparameter search for all hyperparameter combinations.
* We correct the manuscript to address the clarity issues raised by reviewers.

---

### Decision · Program_Chairs · 2023-01-20

**Decision:**

Accept: poster

**Justification For Why Not Higher Score:**

The paper was deemed interesting for the community even if it had several issues, yet well addressed by the authors.
I decided for poster just because of these several issues, not allowing to reach the full agreement among reviewers.

**Justification For Why Not Lower Score:**

The paper could have been rejected but the majority of reviewers, with good motivations and going into details, expresses the will to accept the paper, and the only reviewer slightly negative (5) declared not to be much expert on this topic (3).

**Metareview: Summary, Strengths And Weaknesses:**

This paper was evaluated by 4 reviewers who have appreciated the work but also have raised a lot of concerns.
Their issues were mostly very well addressed by the authors, also deepening the discussion more times with those reviewers who wanted more explanation, and adding further results whenever requested.
After the round of evaluation and authors feedback, the final ratings were 6, 5, 6, 8, hence, only a reviewer remained below threshold, whereas the others overall evaluated the paper positively. Reviewers also discussed separately to contrast overall pros and cons of the method. Comments and discussions revolved around the justifications of the results, which were not always consistent across all datasets, and the additional burden generated by the use of the memory, whose advantage was sometimes not so evident when considering the additional computational cost.
The result is that this paper, even if imperfect, is considered valuable for the community and then accepted to ICLR 23.
AC and reviewers recommend authors to address and include in the final version all the remarks and discussions contained in the reviews, and to clearly and fairly present their method making it clear about assumptions and limitations of their approach, without overclaiming the goodness of it over other state-of-the-art methods.


**Note From Pc:**

if the above contains the word "oral" or "spotlight" please see: "oral" presentation means -> notable-top-5% and "spotlight" means -> notable-top-25%. As stated in our emails, we are disassociating presentation type from AC recommendations

**Summary Of Ac-Reviewer Meeting:**

N/A